# An Enhanced Team-Oriented Swarm Optimization Algorithm (ETOSO) for Robust and Efficient High-Dimensional Search

**DOI:** 10.3390/biomimetics10040222

**Published:** 2025-04-03

**Authors:** Adel BenAbdennour

**Affiliations:** College of Engineering, Islamic University of Madinah, Madinah 42351, Saudi Arabia; aabdennour@iu.edu.sa

**Keywords:** enhanced team-oriented swarm optimization, nature-inspired algorithms, optimization algorithms, high-dimensional search spaces, exploration and exploitation, benchmarks

## Abstract

This paper introduces the Enhanced Team-Oriented Swarm Optimization (ETOSO) algorithm, a novel refinement of the Team-Oriented Swarm Optimization (TOSO) algorithm aimed at addressing the stagnation problem commonly encountered in nature-inspired optimization approaches. ETOSO enhances TOSO by integrating innovative strategies for exploration and exploitation, resulting in a simplified algorithm that demonstrates superior performance across a broad spectrum of benchmark functions, particularly in high-dimensional search spaces. A comprehensive comparative evaluation and statistical tests against 26 established nature-inspired optimization algorithms (NIOAs) across 15 benchmark functions and dimensions (D = 2, 5, 10, 30, 50, 100, 200) confirm ETOSO’s superiority relative to solution accuracy, convergence speed, computational complexity, and consistency.

## 1. Introduction

Optimization plays a crucial role across diverse scientific and engineering disciplines. From machine learning and artificial intelligence to logistics and finance, the ability to efficiently find optimal solutions is important. However, many real-world optimization problems are characterized by high dimensionality and complex landscapes, posing significant challenges to traditional optimization techniques. Nature-inspired optimization algorithms (NIOAs) have emerged as a powerful paradigm for tackling such complex optimization problems. Inspired by natural processes such as swarm behavior and physical phenomena, NIOAs offer robust and adaptable search strategies [1,2,3]. Algorithms like Particle Swarm Optimization (PSO) have achieved notable success across a wide range of applications, often outperforming traditional gradient-based methods in complex, non-convex search spaces. Their inherent parallelism, stochastic nature, and ability to escape local optima make them particularly well suited for challenging optimization tasks.

PSO, inspired by the social foraging behavior of birds and fish, has gained widespread popularity due to its simplicity and effectiveness. It simulates a swarm of particles exploring the search space, with each particle adjusting its trajectory based on its own best-found solution and the best solution found by the entire swarm. While PSO has proven effective in numerous applications, it is susceptible to suboptimal convergence, particularly in complex, multimodal landscapes, where the algorithm can become trapped in local optima, hindering the discovery of the true global optimum.

To address the limitations of the standard PSO, the Team-Oriented Swarm Optimization algorithm (TOSO) was introduced [4]. TOSO incorporates a team-based approach, dividing the swarm into two distinct teams: explorers and exploiters. The explorers navigate a wider area within the search space while the exploiters refine the current best position. This way, the diversity of the population and the robustness of the search process are maintained. This division of labor promotes a more balanced exploration–exploitation trade-off, diminishing the risk of early stagnation and improving the overall search efficiency.

While TOSO represents a significant advancement over the standard PSO, with remarkable performance in optimizing a wide range of benchmark functions, it faces challenges in certain scenarios, particularly in parameter tuning, which can significantly influence performance. This limitation motivates the development of an Enhanced Team-Oriented Swarm Optimization algorithm (ETOSO). ETOSO is a novel enhancement of TOSO that incorporates several key improvements to further enhance exploration, exploitation, simplicity, and overall optimization performance. ETOSO builds upon the team-based structure of TOSO but introduces mechanisms to make the algorithm parameter-free, simplifying implementation and enhancing robustness. These enhancements are designed to address the limitations of TOSO and achieve superior performance in complex optimization problems. The main contributions of this paper are as follows:The development of ETOSO, an enhanced version of TOSO with improved exploration and exploitation strategies and a parameter-free design;A comprehensive experimental evaluation of ETOSO on a diverse suite of 15 benchmark functions, demonstrating its superior performance compared to TOSO and 25 state-of-the-art NIOAs;A detailed computational and statistical analysis of the ETOSO algorithm, providing insights into its behavior and performance characteristics.

The paper is organized as follows: Section 2 reviews the related work. Section 3 provides a detailed background on TOSO, including its mathematical formulations and underlying principles. Section 4 describes the proposed ETOSO algorithm, highlighting the key enhancements and their rationale. Section 5 presents the experimental setup and benchmark functions used for evaluation. Section 6 compares ETOSO with TOSO and other state-of-the-art algorithms. It also provides a statistical and computational complexity analysis. The discussion and limitations are presented in Section 7. Finally, Section 8 concludes the paper and outlines future research directions.

## 2. Related Work

NIOAs have emerged as powerful paradigms for addressing complex optimization challenges, drawing inspiration from natural phenomena to navigate complex solution landscapes. These algorithms offer a dynamic and adaptable approach, proving particularly valuable in tackling nonlinear and high-dimensional problems where traditional optimization techniques often fail. However, while NIOAs present a compelling alternative, their original formulations are not without inherent limitations. Refining NIOAs’ applicability requires understanding the limitations related to parameter sensitivity, exploration–exploitation balance, constraint handling, scalability, and generalizability.

One of the most common challenges across a spectrum of NIOAs is the strong dependence on precise parameter tuning. Algorithms such as the Bat Algorithm (BAT) [5,6], Bees Algorithm (BEE) [7], Flower Pollination Algorithm (FPA) [8], Sine Cosine Algorithm (SCA) [9], Whale Optimization Algorithm (WOA) [10], and Cuckoo Search (CS) [11] can sometimes be particularly susceptible to this issue. The performance of these algorithms may vary dramatically with slight changes in control parameters, demanding careful adjustments to achieve optimal results. This sensitivity may complicate their application in real-world scenarios and may raise questions about their robustness and generalizability.

Furthermore, a significant proportion of original NIOAs may exhibit an inherent difficulty in maintaining a balanced equilibrium between exploration and exploitation. Algorithms like the Butterfly Optimization Algorithm (BOA) [12], Elephant Herding Optimization (EHO) [13], Firefly Algorithm (FA) [14,15], Grasshopper Optimization Algorithm (GOA) [16], Harris Hawks Optimization (HHO) [17], Moth–Flame Optimization (MFO) [18], Slime Mold Algorithm (SMA) [19], and Salp Swarm Algorithm (SSA) [20,21] can frequently struggle to navigate this trade-off effectively in some contexts. Over-emphasis on exploration can lead to inefficient searches, while excessive exploitation can result in premature convergence and trap in local optima. This underscores the need for more sophisticated search strategies that dynamically adapt to the problem landscape.

Moreover, the practical deployment of a number of NIOAs is frequently restricted by limitations in constraint handling and scalability. Algorithms such as the Crow Search Algorithm (CSA) [22], Grey Wolf Optimizer (GWO) [23], Teaching–Learning-Based Optimization (TLBO) [24], Flow Direction Algorithm (FDA) [25], Gravity Search Algorithm (GSA) [26], Raven Roost Optimization (RRO) [27,28], Monkey King Algorithm (MKA) [29,30,31,32], and Remora Optimization Algorithm (ROA) [33,34] may demonstrate difficulties in effectively managing constraints and scaling to high-dimensional problems in certain applications. These limitations can pose significant challenges in real-world applications where constraints are prevalent, and problem dimensions are large, often leading to increased computational complexity and diminished solution quality.

Finally, a subset of NIOAs faces concerns regarding their theoretical foundations and generalizability. Differential Evolution (DE) [35], for instance, lacks comprehensive theoretical insights into its convergence properties, while the Prairie Dog Optimization (PDO) algorithm [36] and the Seagull Optimization Algorithm (SOA) [37] require more extensive comparative analyses to validate their performance across diverse problem domains. This lack of robust theoretical studies and empirical validation may limit their broader applicability.

In response to the various limitations inherent to some original NIOAs, researchers have dedicated considerable effort to developing enhancements and hybridizations, aiming to augment performance and expand applicability. To address parameter sensitivity, a prevalent strategy has been the implementation of adaptive parameter control. Adaptive versions of the Bat Algorithm and FPA, for example, dynamically adjust parameters based on search progress, thereby enhancing robustness and accelerating convergence. Specific implementations, such as the dynamic adaptation of Levy flight step size in CS and variable step size in Firefly FA [38], demonstrate this approach. These adaptive mechanisms seek to reduce the reliance on manual parameter tuning, fostering greater autonomy and adaptability within the algorithms, though their efficacy remains dependent on the specific problem landscape.

To address the persistent challenges in achieving a balanced equilibrium between exploration and exploitation, hybrid approaches have gained significant traction. These methods combine the strengths of other algorithms to optimize search efficacy. Example instances of such an approach include hybridizations of SSA with PSO [39], SCA with TLBO [40], and GWO with CS [41]. Furthermore, enhancements to FPA [42] and TLBO [43] have focused on refining the balance between exploration and exploitation, thereby improving convergence and solution quality. However, the increased complexity of these hybrid approaches can raise concerns about computational efficiency.

Advancements in constraint handling and scalability have been realized through various modifications. Enhanced versions of CSA and TLBO have incorporated strategies to manage infeasible solutions more effectively, thereby improving performance in constrained optimization problems. For high-dimensional problems, enhancements to HHO [44,45,46,47] and CSA [48] have yielded promising results through multi-strategy enhancements and improved search mechanisms. Similarly, enhanced BEE algorithms have integrated Deb’s rules [49] to improve constraint-handling capabilities. These modifications demonstrate a commitment to expanding the applicability of NIOAs to real-world problems.

Furthermore, efforts have also been directed toward enhancing the theoretical foundations and generalizability of NIOAs. Modifications to MFO [50], SCA [51], SOA [52], and GWO [53,54] have focused on improving convergence, accuracy, and general applicability across diverse optimization landscapes. These enhancements aim to solidify the theoretical underpinnings of NIOAs and broaden their practical deployment, though standardized benchmarking remains a critical need for rigorous performance evaluation.

Despite these advancements, challenges are still being investigated. The increasing complexity of hybrid approaches raises concerns regarding computational efficiency and practical applicability. Consequently, future research should prioritize the development of robust, efficient, and generalizable NIOAs characterized by reduced parameter sensitivity and transparent tuning methodologies. There remains a pressing need for algorithms that exhibit diminished reliance on precise parameter adjustments or at least provide a clear and methodical tuning approach while delivering superior performance across a spectrum of optimization problems, and this is an area of ongoing research.

## 3. The TOSO Algorithm

The original PSO is a simple population-based optimization algorithm inspired by the social behavior of bird flocks. In a D-dimensional search space, the position and velocity of the *i*th particle in the *d*th dimension are updated at each iteration using the following equations [55]:(1)$$v_{i,d}=\left(w\times v_{i,d}\right)+c_{1}\times r_{1}\times \left({pbest}_{i,d}-x_{i,d}\right)+c_{2}\times r_{2}\times \left({gbest}_{d}-x_{i,d}\right)$$(2)$$x_{i,d}=x_{i,d}+v_{i,d}$$
where
$x_{i,d}$: Position of the *i*th particle in the *d*th dimension;$v_{i,d}$: Velocity of the *i*th particle in the *d*th dimension;w: Inertia weight, controlling the influence of the previous velocity;$c_{1}$, $c_{2}$: Acceleration constants, influencing the attraction toward personal and global best positions;${pbest}_{i,d}$: Personal best position of the *i*th particle in the *d*th dimension;${gbest}_{d}$: Global best position of the swarm in the *d*th dimension;*r*_1_ and *r*_2_ are random numbers.

PSO balances exploration and exploitation through the careful selection of the acceleration constants ($c_{1}$ and $c_{2}$) and the inertia weight (*w*).

TOSO is an innovative enhanced version of PSO that aims to improve its performance by decoupling exploration and exploitation. In PSO, a single swarm explores the solution space. In contrast, TOSO divides the swarm into two teams: an exploration team responsible for discovering new areas of the search space and an exploitation team dedicated to refining potential solutions. The only information shared between the teams is the current best position. This decoupling allows TOSO to balance exploration and exploitation more effectively, resulting in performance enhancement in navigating complex search spaces, particularly in multimodal problems where multiple optimal solutions may exist. This enhancement allowed for a much-improved performance over a large number of benchmarks, even for very high dimensions [4].

### 3.1. TOSO Exploration Team

The exploration team aims to discover promising new regions within the search space. The explorers are guided by the local best (*lbest*) model instead of the PSO global (*gbest*) model. This enhances exploration by focusing on the best local positions, preventing rapid convergence, and maintaining diversity within the team. TOSO also uses the ring topology for neighbor selection instead of the PSO star topology for the same reason. Moreover, TOSO does not use a velocity update, and the new position is directly determined from the previous one. Given these modifications, the position of each explorer is, hence, updated according to the following equation:(3)$$x_{i,d}=\alpha \times r_{1,d}\times \left({lbest}_{i,d}-x_{i,d}\right),$$
where
$x_{i,d}$ represents the new position of the *i*th explorer;*lbest* is the best position found by the explorer’s local neighbors within a ring topology;*r*_1*,d*_ is a uniform random number between 0 and 1, different for each dimension;

The exploration factor (α) is dynamically adjusted to balance exploration and exploitation. It is calculated as follows:
(4)$$\alpha =\left|\frac{min(fitness)}{1+max(fitness)}\right|,$$
where *fitness* is the vector containing the fitness values of all explorers. This dynamic adjustment of the exploration factor ensures that the team maintains a balance between exploiting promising regions and exploring new areas of the search space. To prevent premature convergence and maintain diversity within the exploration team, TOSO applies a random mutation operator to explorers with a predefined probability *p_m_*. The rebirth is achieved using the following:(5)$$x_{i,d}=R_{min,d}+r_{2,d}\times \left(R_{max,d}-R_{min,d}\right)$$
where
*R_min_* and *R_max_* represent the lower and upper bounds of the search space, respectively;*r*_2*,d*_ is a uniform random number between 0 and 1, different for each dimension.

### 3.2. TOSO Exploitation Team

The exploitation team refines the search around the current best solution. Instead of moving toward the best position, each exploiter is initialized at the best solution and then undergoes small, controlled movements within a localized region. The magnitude of these movements is determined by an exploitation factor (*β_j_*), which is individually assigned to each exploiter based on its relative performance within the exploitation team. The position update for each exploiter is given by the following:(6)$$x_{j}=x_{j}+M_{j,d}$$(7)$$M_{j,d}=\beta_{j}\times r_{3}\times \left(R_{max,d}-R_{min,d}\right)$$
where *r*_3_ is a random number drawn from a standard normal distribution (to favor exploiting closer proximities) and *M_j,d_* is a small perturbation motion for dimension *d.* Denoting *k* to be the particle’s relative position within the swarm, determined by its fitness level, and *ps* being the swarm size, the exploitation function is given by the following:(8)$$\beta \left(k\right)=\gamma_{1}+\gamma_{2}\times \frac{\left(e^{\left(\frac{\gamma_{3}\left(k-1\right)}{0.5*ps-1}\right)}-1\right)}{\left(e^{\left(\gamma_{3}\right)}-1\right)},$$
where *γ*_1_, *γ*_2_, and *γ*_3_ are constants that control the search radius of the exploiters. TOSO makes the best particle exploit within 0.01% of the range and the worst one within 50% of the range, relative to their current position. Under these conditions, the parameters are determined to be [55] *γ*_1_ = 0.001, *γ*_2_ = 0.499, and *γ*_3_ = 2.

## 4. ETOSO Algorithm

ETOSO incorporates several key enhancements to TOSO that are aimed at improving its exploration and exploitation capabilities. These enhancements include linear weight increment for exploitation, the indirect refined neighbor selection, the complete removal of the rebirth step of explorers, and avoiding any need for parameter settings. With these enhancements, ETOSO becomes a simple and parameter-free algorithm while maintaining a competitive performance.

### 4.1. Linear Weight Increment for Exploitation

TOSO employs an exponential weight increment for exploiters, using three parameters as given by (8). The exponential weighting strategy in TOSO, besides the inconvenience of having to use three parameters, can lead to significant disadvantages. While the weights guide exploiters toward optimizing the global best solution, the nonlinear nature of exponential weighting makes the algorithm overly sensitive to changes in fitness rankings, resulting in abrupt alterations in the weighted influence applied to the positions of the exploiters. This behavior may prompt exploiters to converge too quickly to local optima, thereby restricting their capacity to adaptively navigate the search space and explore alternative promising solutions.

Alternatively, ETOSO proposes a much simpler linear weighting strategy for the exploitation function given by the following:(9)$$\beta \left(k\right)=\gamma_{1}(k-1)+\gamma_{2}$$

The slope and bias are selected so that the starting and ending weights of ETOSO are similar to that of TOSO, as shown in Figure 1 for a population equal to 30. For any population *ps*, the values of $\gamma_{1}$ and $\gamma_{2}$ are given by the following:(10)$$\gamma_{1}=\frac{0.996}{ps-2},\mathrm{and}\;\gamma_{2}=0.001$$

The linear behavior, aside from being simple and less computationally demanding, allows for gradual and consistent adjustments to the weights, promoting smoother movements and reducing potential oscillations among exploiters. While both strategies utilize particle rankings, the linear approach ensures that exploiters can exploit good solutions without the erratic fluctuations that can affect convergence. A linear increase in weights provides a more controlled and progressive shift and ensures that the exploitation influence grows gradually, allowing the algorithm to explore the search space effectively while progressively focusing on promising regions. This approach enables ETOSO to maintain a balanced and effective search process, allowing for better adaptation to changes in the fitness landscape and enhancing the algorithm’s overall robustness and performance.

### 4.2. Dynamic Exploration Enhancement

TOSO incorporates randomization through a mutation probability *p_m_*. While this introduces some degree of exploration, the level of exploration is heavily dependent on the predetermined value of *p_m_*, which may not be optimal across different problem contexts. The authors of TOSO themselves acknowledged the lack of justification behind their choice of the value of *p_m_*. In reality, a fixed mutation probability may not adequately serve all optimization scenarios, as higher mutation rates might be required to effectively escape local optima, while lower rates could help maintain focus on promising regions.

However, after examining the effect of the mutation of the explorers, it was found that it does not really have any useful addition to the process. In fact, it adds to the complexity of TOSO, increases the average speed, and makes an additional variable to tune without significant benefit. Therefore, ETOSO employs only the basic exploration given by (3) with no rebirth. This simplifies the algorithm with even some potential improvements in performance, speed, and consistency.

### 4.3. Indirect Refined Neighbor Selection

While both TOSO and ETOSO engage in neighbor selection for each explorer based on their indices (including previous, current, and next explorers), ETOSO’s incorporation of a linear weight increment for exploitation can indirectly enhance the influence of neighbor interactions. The gradual adjustment of weights provides a more stable and predictable movement pattern for exploiters, leading to a smoother transition between exploration and exploitation phases. Consequently, this consistency may facilitate more effective utilization of information from neighboring explorers during the exploration process. By capitalizing on knowledge from nearby particle performances, ETOSO can achieve more informed search directions, resulting in improved exploration efficiency within the optimization task. The pseudocode for the ETOSO algorithm is presented in Figure 2.

## 5. Experimental Setup

The experimental setup was designed to rigorously evaluate the performance of the proposed algorithm. This section describes the benchmark functions, the comparative algorithms, and the test configuration used in the study.

### 5.1. Benchmark Functions

A diverse set of benchmark functions was selected in accordance with CEC guidelines [56,57] to validate the proposed algorithm. These functions include unimodal functions (f1: Sphere, f2: Zakharov, f3: Sum Squares), multimodal functions (f4: Schwefel’s Problem 2.22, f5: Schwefel’s Problem 2.26, f6: Rosenbrock, f7: Ackley, f8: Rastrigin, f9: Griewank, f10: Bent Cigar), shifted functions (f11: Shifted Ackley, f12: Shifted Rosenbrock, f13: Shifted Rastrigin, f14: Shifted Griewank), and rotated functions (f15: Rotated Griewank). The equations for all benchmark functions are provided in Table 1. Unimodal functions (f1–f3) are primarily used to evaluate the exploitation capability of the algorithm, while multimodal functions (f4–f10) assess its exploration effectiveness and ability to avoid local optima. The shifted (f11–f14) and rotated (f15) functions introduce additional complexity by altering the location and orientation of the global optimum, making them more challenging and realistic for evaluating algorithm performance.

To provide a visual understanding of the functions, 3D plots of all 15 benchmark functions are included in Figure 3. These plots illustrate the unique characteristics of each function, such as the number of local optima, symmetry, and overall landscape complexity.

### 5.2. Comparative Algorithms and Parameter Settings

ETOSO was compared against a wide range of state-of-the-art optimization algorithms, namely BAT, BEE, BOA, CSA, CS, DE, EHO, FA, FDA, FPA, GOA, GSA, GWO, HHO, MFO, MKA, PDO, ROA, RRO, SCA, SOA, SMA, SSA, TLBO, and WOA. These algorithms were selected to provide a comprehensive comparison across different optimization paradigms, including swarm intelligence, evolutionary algorithms, and physics-inspired methods. The optimal parameter settings for each algorithm, obtained from the literature, are summarized in Table 2. These settings were carefully chosen to ensure fair and accurate comparisons.

### 5.3. Test Configuration

The experiments were conducted using MATLAB 2024 (version R2024a) on a system equipped with an Intel (R) Core (TM) Ultra 9 185 H processor (2.30 GHz), 32.0 GB RAM, and a 64-bit operating system. All algorithms used a population size of 30 individuals (except for MKA, which uses 10 due to many additional internal iterations). The maximum number of function evaluations (FEs) for each dimension (D) was set to 5000 × D, following standard CEC guidelines [56,57]. All results were averaged over 25 independent runs for each algorithm and benchmark function to ensure statistical robustness evaluation. Each benchmark function was evaluated across multiple dimensions (D = 2, 5, 10, 30, 50, 100, and 200). To facilitate the reproduction of the results, the ETOSO algorithm code is available at https://github.com/adel468/ETOSO.

## 6. Experimental Results

In this section, we plan to evaluate the performance of the ETOSO algorithm through four different approaches. The first approach will compare ETOSO with TOSO to identify any potential improvements or differences that ETOSO presents. The second approach expands this analysis by contrasting ETOSO with 25 other algorithms, providing insights into its relative strengths and possible applications. The third approach focuses on analyzing statistical significance and sensitivity to population size. Finally, the last approach will examine the computational complexity. All these evaluations are designed to assess ETOSO’s effectiveness and its significance within the broader context of algorithmic approaches.

### 6.1. Comparative Analysis of ETOSO and TOSO

Figure 4 shows the convergence behavior of the two algorithms for sample benchmark functions with D = 30. Table 3 and Table 4 present the results of the experimental evaluation for dimensions D = 2, 5, 10, 30, 50, 100, and 200. Each table provides the best performance, mean performance, performance standard deviation, and average speed (over 25 replications).

The results show that both algorithms demonstrate strong performance in simpler unimodal functions, consistently achieving optimal values of zero across various dimensions. This indicates their capacity for effective exploration and exploitation in landscapes characterized by a single peak. In contrast, the performance in multimodal functions reveals that while both algorithms maintain a comparable level of effectiveness, ETOSO occasionally outperforms TOSO in speed, especially with increasing dimensions. This suggests that ETOSO may possess enhanced adaptability to complex solution spaces with multiple optima.

Notably, in challenging functions such as the shifted and rotated benchmarks, both algorithms demonstrate their capacity to handle various complexities. However, ETOSO shows significantly lower variability in its results across higher dimensions, indicating greater stability and reliability compared to TOSO. The analysis underscores ETOSO’s performance in tackling complex problems with multiple local optima and deceptive features, notably in higher dimensions (D = 30, D = 50, D = 100, and D = 200). It highlights relatively faster convergence rates and robustness, particularly in navigating rugged landscapes, while maintaining a lower standard deviation compared to TOSO. Overall, while both algorithms managed simpler problems, ETOSO emerges as a robust and versatile option for complex, high-dimensional optimization tasks, balancing considerations of speed, stability, and adaptability in diverse solution landscapes exceptionally well.

The key takeaway from this first comparative test is that ETOSO performs as well as TOSO and often outperforms it, with greater consistency, in various scenarios. This enhanced performance is attained through a simpler structure and without the necessity for parameter tuning. Such improvements position ETOSO as a strong contender for optimization problems, which can be validated by benchmarking it against recently developed algorithms.

### 6.2. Benchmarking ETOSO Against Other Algorithms

This subsection presents the results of the benchmarking study. Figure 5 displays the convergence plots for the 25 algorithms across 25 replications, specifically for dimensionality D = 30. This figure highlights the sample performance dynamics of each algorithm for one multimodal function (f7) and one rotated function (f15). It is important to note that the FDA algorithm has been excluded from this figure due to its considerably slower convergence, which would distort the comparative analysis of the other algorithms.

Table 5 and Table 6 provide comprehensive results for all algorithms for D = 5, evaluated over 25 replications with a total of 5000 FEs per dimension. Table 5 focuses on functions f1 through f8, while Table 6 covers functions f9 through f15. Furthermore, Table 7 ranks all algorithms based on their average performances for dimensionalities D = 2, D = 5, and D = 10, with a focus on identifying the top-performing algorithms for further analysis, particularly in higher-dimensional scenarios. This ranking aims to distill key insights regarding the strengths of ETOSO and its competitors, thus guiding subsequent investigations into their behaviors under more complex conditions. Together, these results will provide a clearer understanding of ETOSO’s capabilities and its positioning within the broader algorithmic landscape.

A significant observation was the performance degradation of many algorithms as the dimensionality of the problem increased. This highlights the inherent challenge of effectively exploring and exploiting the search space in higher-dimensional problems. The search space expands exponentially with increasing dimensionality, making it more difficult for algorithms to locate the global optimum. Based on the results, the 25 algorithms can be grouped into three tiers: top performers, average performers, and low performers.

The first tier includes ETOSO, FPA, DE, GWO, TLBO, ROA, and SCA. This tier encompasses algorithms that consistently demonstrate high-ranking performance across all dimensions. These algorithms exhibited a strong balance of exploration and exploitation capabilities, effectively navigating the search space and converging toward optimal or near-optimal solutions. ETOSO, in particular, consistently achieved top rankings, highlighting its robustness and adaptability to varying problem complexities.

The second tier includes WOA, MFO, HHO, MKA, FDA, PDO, BAT, EHO, RRO, BEE, BOA, FA, and GOA. This tier includes algorithms that exhibited moderate performance across different dimensions. While they achieved reasonable results, their performance might degrade in higher-dimensional problems, suggesting potential limitations in their capacity to thoroughly explore and exploit the search space in more complex scenarios.

The third tier includes CSA, CS, SSA, GSA, SMA, and SOA. This tier comprises algorithms that consistently ranked among the low performers across all dimensions, indicating significant limitations in their exploration and exploitation capabilities. These algorithms often struggled to escape local optima, leading to suboptimal solutions or even divergence, particularly in higher-dimensional problems.

Based on the rankings in Table 7, we have selected the top 10 performing algorithms for a detailed evaluation of how ETOSO performs against them in more complex scenarios. This assessment will specifically explore ETOSO’s effectiveness and examine any limitations in higher-dimensional and multimodal optimization problems. By comparing its performance to these established algorithms, we aim to highlight ETOSO’s strengths and identify areas for improvement, providing a comprehensive understanding of its competitive position in challenging optimization landscapes.

Table 8 details the evaluation of these algorithms for functions f1 through f8, with a dimensionality of D = 200, offering insights based on 25 replications and 5000D FEs. Similarly, Table 9 presents the evaluation for functions f9 through f15 under the same conditions. These tables collectively provide a deep dive into the capabilities and behavior of the algorithms in high-dimensional settings. Subsequently, Table 10 ranks the top algorithms in terms of their overall performance (P), speed (S), and consistency (C) for dimension D = 30, 50, 100, and 200.

As seen in Table 10, high-dimensional scenarios posed significant challenges for most algorithms, negatively impacting their performance. The complexity associated with these high dimensions emphasizes the need for robust optimization strategies. As the dimensionality of the problems increased, most algorithms exhibited a general decline in performance, with FPA and MFO being particularly vulnerable in these scenarios.

Based on the evaluation, the order of the algorithms from the top performer down is ETOSO, GWO, HHO, ROA, DE, SCA, TLBO, WOA, FPA, MFO, and PDO. ETOSO emerged as the best overall performer, demonstrating exceptional speed and consistency across all dimensions. GWO followed closely with strong performance and good speed, while HHO was noted as the fastest algorithm. ROA exhibited satisfactory performance along with reliable consistency. DE and SCA showed moderate performance levels, with TLBO being slightly less competitive in higher dimensions. WOA also demonstrated reasonable performance, though not as strong as the top contenders. FPA and MFO struggled in complex scenarios, particularly in higher dimensions, and PDO consistently displayed lower performance with reliability issues.

### 6.3. Analysis of Statistical Significance and Population Size Sensitivity

A comprehensive statistical analysis was undertaken to evaluate the performance of ETOSO in comparison to the 10 top-performing algorithms across a range of benchmark functions (f1–f15), particularly focusing on the impact of significantly reducing the population size from 30 to 10 in a high-dimensional problem space (D = 50). The analysis incorporated the Wilcoxon signed-rank test, *p*-values, and Cliff’s Delta to provide a robust assessment of statistical significance and effect size for 25 replications.

The Wilcoxon signed-rank test, a non-parametric statistical hypothesis test, was employed to determine whether there were statistically significant differences in the performance of ETOSO compared to the other algorithms. The test assesses whether pairs of observations (in this case, performance metrics of two algorithms on the same function) are selected from the same distribution. The resulting *p*-values, presented in Table 11, indicate the probability of observing the obtained results (or more extreme results) if there were no actual differences between the algorithms. A *p*-value below a predetermined significance level (α = 0.05) suggests that the null hypothesis (no difference) can be rejected, indicating a statistically significant difference in performance. In this study, statistically significant differences (*p* ≤ 0.05) were observed in 74% of the comparisons between ETOSO and the other algorithms across the benchmark functions, indicating that ETOSO’s performance is not equivalent to the other algorithms for a majority of the tested cases. Specifically, statistically significant differences were observed in the performance of ETOSO compared to DE, FPA, HHO, MFO, and TLBO across all 15 benchmark functions. Very low *p*-values strongly suggest that the performance of ETOSO is significantly different from these algorithms. The Wilcoxon Significance results, shown in Table 12, further reinforce this, with values of 1 consistently showing for these five algorithms when compared to ETOSO, confirming the statistical significance.

Complementing the *p*-values, Cliff’s Delta, shown in Table 13, was used to quantify the effect size, providing a measure of the magnitude of the difference between the algorithms. A negative Cliff’s Delta value indicates that the second algorithm (the comparative algorithm) is stochastically dominated by the first algorithm (ETOSO). In other words, a negative Cliff’s Delta signifies that ETOSO consistently outperformed the other algorithm. Notably, Cliff’s Delta values approaching −1 were observed for comparisons between ETOSO and DE, FPA, HHO, MFO, and TLBO across all 15 functions. These values indicate a substantial effect size, demonstrating that ETOSO’s performance was consistently and significantly superior to these algorithms. Specifically, a Cliff’s Delta of −1 indicates that ETOSO stochastically dominates the other algorithm, meaning that ETOSO’s values are consistently smaller than the other algorithm’s values across all paired comparisons.

In comparisons with GWO, PDO, ROA, SCA, and WOA, the statistical analysis reveals that ETOSO demonstrates neither a statistically significant advantage nor disadvantage across all benchmark functions. To provide further differentiation of algorithm performance where the statistical analysis identified comparable results, the detailed performance metrics in Table 14 will be utilized for further assessment. These metrics are defined as follows:Number of Perfect Hits: This metric counts the instances in which an algorithm solution precisely matches the known optimal solution;Number of Times Closest to Minimum: This metric assesses an algorithm’s ability to achieve the average performance closer than any other compared algorithm to the known minimum across benchmark functions, indicating superior near-optimal convergence;Number of Std Dev ≤ Threshold: This metric reflects the algorithm’s consistency by counting instances where the performance standard deviation meets a threshold, defined as 1e-6 multiplied by the absolute known minimum for non-zero minima and 1e-6 for zero minima;Average Normalized Error (NAE): This metric is calculated as the mean of individual normalized errors, derived from the absolute difference between average performance and known minimum, divided by the absolute known minimum or 1 for zero minima, providing a scale-invariant error measure;Trimmed NAE (TNAE): This metric is the Average Normalized Error (NAE) with the outlier from each algorithm result removed. This is used to provide a more accurate and fair representation of performance by ensuring that comparisons reflect typical results rather than extreme cases;Average Speed: This metric represents the mean execution time in seconds across all benchmarks, indicating computational efficiency.

Analyzing the results presented in Table 14, ETOSO demonstrates strong performance across multiple metrics. ETOSO achieved a high number of perfect hits (9), tying for first place with several other algorithms. Furthermore, it exhibited the highest number of times closest to the minimum (9), showcasing its robust convergence behavior. ETOSO also demonstrated the highest level of consistency, with 14 instances of standard deviation meeting the predefined threshold. ETOSO’s Average Normalized Error (NAE) of 1.87 × 10^3^ (4.03 if the outlier is removed for each algorithm) is lower than the majority of the comparative algorithms, demonstrating its ability to find solutions that are close to the optimal values. Additionally, ETOSO exhibited a very low average speed (0.852), ranking second only to HHO, indicating its high computational efficiency and consistent execution time.

These results, combined with the statistical significance demonstrated in the preceding analysis, highlight ETOSO’s superior performance across a range of benchmark functions. Importantly, ETOSO’s performance was consistent even with a significant reduction in population size (from 30 to 10) in a high-dimensional problem space (D = 50). This insensitivity to changes in population size demonstrates ETOSO’s robustness and scalability, making it a powerful and reliable optimization algorithm for challenging optimization problems.

### 6.4. Computational Complexity and Overhead Analysis

This section presents a comprehensive analysis of the computational complexity and practical overhead inherent in the optimization algorithms under study. While original research papers often lack formal complexity analyses, we derive these complexities based on the algorithms’ structural operations and scaling properties with problem dimensionality (D), population size (ps), and function evaluations (FE). The computational complexity of these algorithms is fundamentally determined by FE, D, and, for ETOSO, ps. All algorithms, except ETOSO, exhibit a complexity of O(FE⋅D), reflecting the positional updates in D-dimensional space. ETOSO, in contrast, demonstrates a complexity of O[FE⋅(D + log (ps)], where the D term arises from positional updates and the log (ps) term stems from sorting operations for neighbor selection. In practical terms, the log (ps) term in ETOSO has a minimal impact on scalability for typical population sizes, particularly in high-dimensional problems where D significantly outweighs log (ps). For instance, with ps = 30, log ps ≈ 3.4, which is negligible compared to D = 500. In this case, the D term dominates ETOSO’s complexity, making the log (ps) term insignificant. However, for ps = 100 and D = 10, log (ps) ≈ 4.6, which becomes more relevant because it is nearly half the size of D. In such lower-dimensional problems, the log (ps) term contributes more significantly to the overall complexity. Nevertheless, for large-dimension optimization scenarios, the D term continues to dominate ETOSO’s complexity, rendering the log (ps) term relatively minor.

Beyond theoretical complexity, practical overhead from hidden operations and constant factors significantly influence algorithm performance. These hidden operations include computationally expensive mathematical functions (trigonometric, exponential, logarithmic), sorting, neighbor selection, random number generation, and conditional logic. Based on code analysis, algorithms can be categorized by overhead, as shown in Table 15.

This direct correlation between overhead classification and algorithm performance, despite similar O(FE⋅D) complexities for all algorithms other than ETOSO, underscores the importance of considering practical overhead alongside theoretical complexity. Algorithms with low overhead, such as HHO, exhibit faster performance, while those with high overhead, like FPA and ROA, are slower. ETOSO’s slightly higher theoretical complexity due to the log (ps) term is mitigated by its efficient implementation. This analysis reinforces that Big O complexity alone does not fully explain observed performance differences, highlighting the necessity of assessing practical overhead for accurate algorithm evaluation. In fact, the empirical results for average speed given in the last column of Table 14 confirm what is in Table 15.

## 7. Discussion and Limitations

ETOSO, with a much simpler structure, presents significant advancements over its predecessor, TOSO, as demonstrated by an extensive benchmarking study against 25 competitive algorithms. The empirical evaluations across diverse benchmark functions suggest that ETOSO shows improved performance parameters and may serve as a strong contender in the optimization landscape.

Initial comparisons between TOSO and ETOSO reveal that both algorithms perform well on simpler unimodal functions, such as the Sphere and Zakharov benchmarks, consistently achieving optimal values of zero across all tested dimensions. However, when transitioning to multimodal functions like Schwefel’s Problem 2.26 and Rosenbrock, a significant distinction emerges. While TOSO demonstrates competent performance, ETOSO sets itself apart with enhanced speed and robustness at dimensions D = 5 and D = 10. This finding underscores the effectiveness of ETOSO’s design improvements, particularly the linear weight increment, which promotes a more stable and gradual exploitation strategy.

The extensive evaluations against 25 leading algorithms further solidify ETOSO’s position as a frontrunner in swarm optimization. The comparative results reveal that ETOSO outperformed competitors such as GWO and HHO, achieving the best overall performance rank among the tested algorithms. Particularly in high-dimensional spaces (D = 30, D = 50, D = 100, and D = 200), ETOSO not only maintained a competitive edge in solution quality but also exhibited faster convergence rates. The performance metrics indicate that ETOSO consistently converges to optimal or near-optimal solutions even in challenging multimodal landscapes characterized by multiple peaks and valleys.

A notable advantage of ETOSO is its robustness; it demonstrated lower variability in results, suggesting that the algorithm is less susceptible to fluctuations and thus offers greater consistency across multiple replications. This reliability is especially significant in practical optimization scenarios where the stability of results can influence decision-making processes. Furthermore, ETOSO’s adaptability is underscored by its performance on shifted and rotated function benchmarks. The algorithm’s ability to generalize across these challenging transformations further confirms its robustness and versatility compared to both TOSO and the competing algorithms. This adaptability points to the potential of ETOSO to effectively tackle a wide variety of real-world optimization problems.

The comprehensive statistical analyses using the Wilcoxon signed-rank test and Cliff’s Delta provided robust evidence of ETOSO’s superior performance, highlighting statistically significant differences in its effectiveness compared to other algorithms across various benchmarks. Additionally, the examination of computational complexity revealed that while ETOSO’s theoretical complexity includes a log (ps) term, its practical efficiency and reduced overhead allow it to maintain competitive execution speeds in high-dimensional optimization tasks.

While ETOSO demonstrates significant performance improvements, several limitations warrant further study. First, a more thorough analysis of ETOSO’s computational complexity is needed to fully assess its scalability and practicality in resource-constrained settings. This analysis should include a detailed comparison of its computational cost (time and memory usage) against other leading algorithms to determine its suitability for various application scenarios and computational budgets. Second, the benchmark functions used, while diverse, may not fully encompass the wide range of complexities inherent in real-world optimization problems. Therefore, evaluating ETOSO’s performance on a broader set of benchmarks, such as those suggested by CEC2017 and CEC2019, in addition to real-world datasets, is crucial to confirm its generalizability and robustness in less-idealized scenarios and to identify any potential limitations in diverse applications.

## 8. Conclusions and Future Research

This study introduces ETOSO as a noteworthy advancement in swarm optimization, demonstrating improvements over its predecessor, TOSO, and performing competitively against a range of 25 algorithms. The enhancements, such as the implementation of a linear weight increment for exploitation and simplification of the algorithm to become parameter-free, have led to faster convergence rates and greater robustness. The benchmarking results and statistical analyses suggest that ETOSO shows encouraging performance across various benchmarks and dimensions, indicating potential advantages in the optimization landscape. Furthermore, ETOSO’s ability to sustain good performance with reduced population sizes implies its adaptability for practical applications, making it a viable option for complex optimization tasks in fields like engineering, finance, and artificial intelligence. However, further investigation into its performance in real-world scenarios and additional problem contexts would be beneficial to fully understand its capabilities and limitations.

Future research should explore the application of ETOSO in dynamic and noisy environments while assessing its performance in real-world settings. Investigating the unique mechanisms behind ETOSO’s success may offer deeper insights into optimization strategies that can benefit various applications. Specifically, applying ETOSO to practical problems is essential, including its effectiveness in engineering design optimization (e.g., optimizing designs under constraints or for multi-objective optimization), supply chain management (enhancing efficiency through inventory and transportation considerations), financial modeling (improving market predictions and portfolio optimization), and machine learning (optimizing hyperparameters). Additionally, conducting a formal theoretical analysis of ETOSO’s convergence characteristics and exploration–exploitation balance, alongside comparisons to other top-performing algorithms, will be crucial for understanding its advantages. Evaluating ETOSO’s robustness under dynamic conditions, such as time-varying objective functions and uncertain parameters, will also help determine its resilience in less predictable real-world scenarios.

## Figures and Tables

**Figure 1 biomimetics-10-00222-f001:**
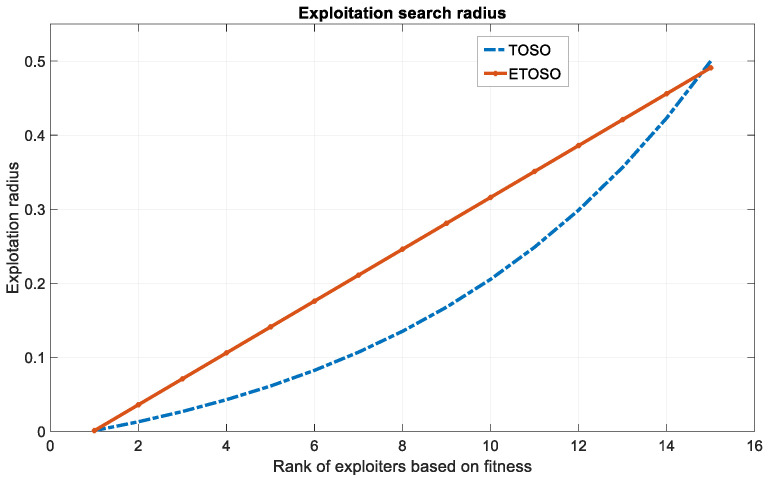
Exploitation function of ETOSO.

**Figure 2 biomimetics-10-00222-f002:**
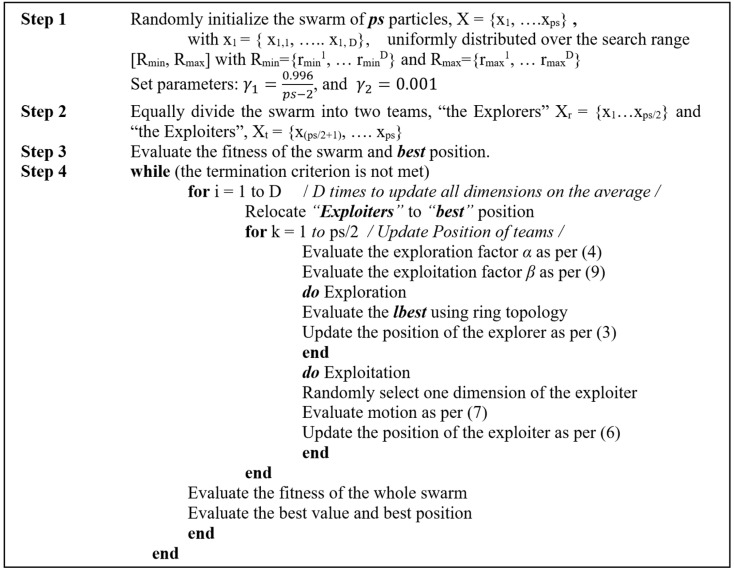
Pseudocode of ETOSO.

**Figure 3 biomimetics-10-00222-f003:**
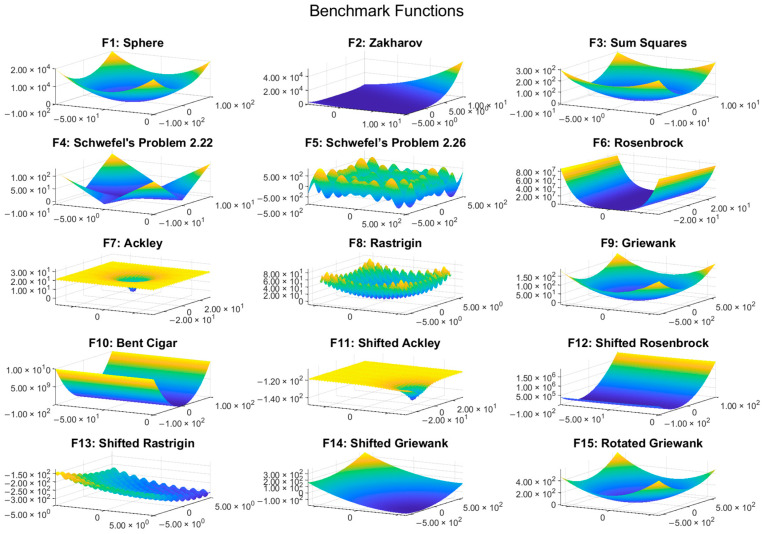
Three-dimensional plots of the benchmark functions.

**Figure 4 biomimetics-10-00222-f004:**
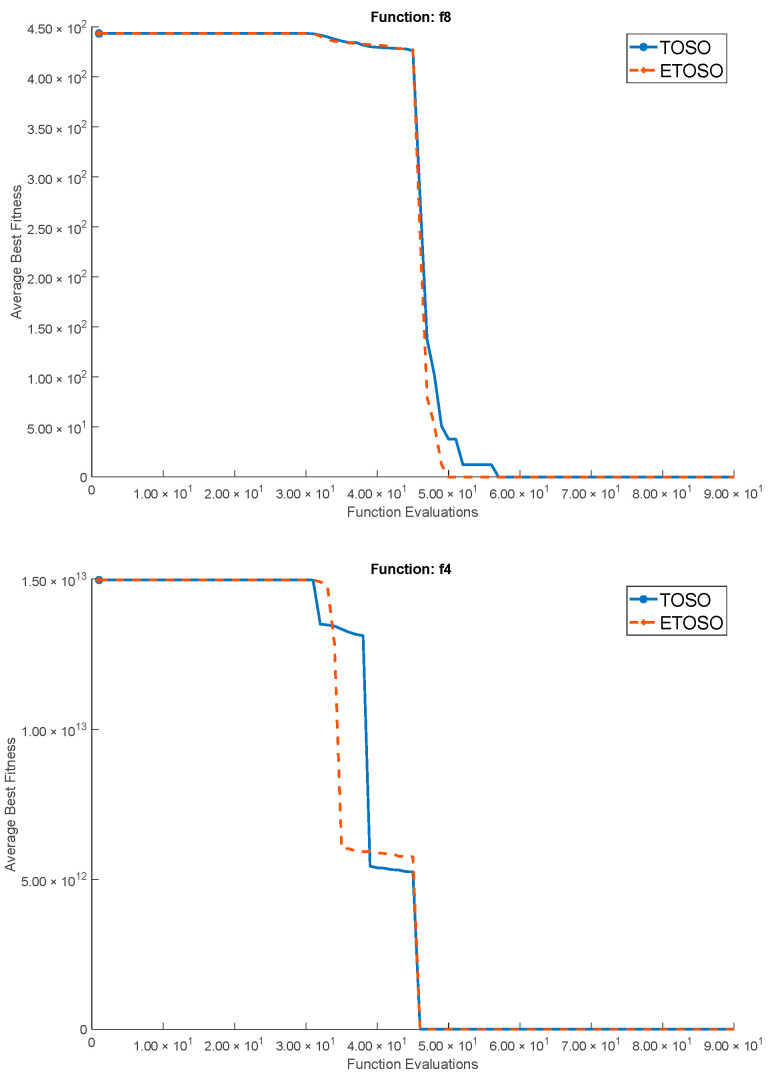
ETOSO vs. TOSO convergence plots for D = 30 and 25 replications.

**Figure 5 biomimetics-10-00222-f005:**
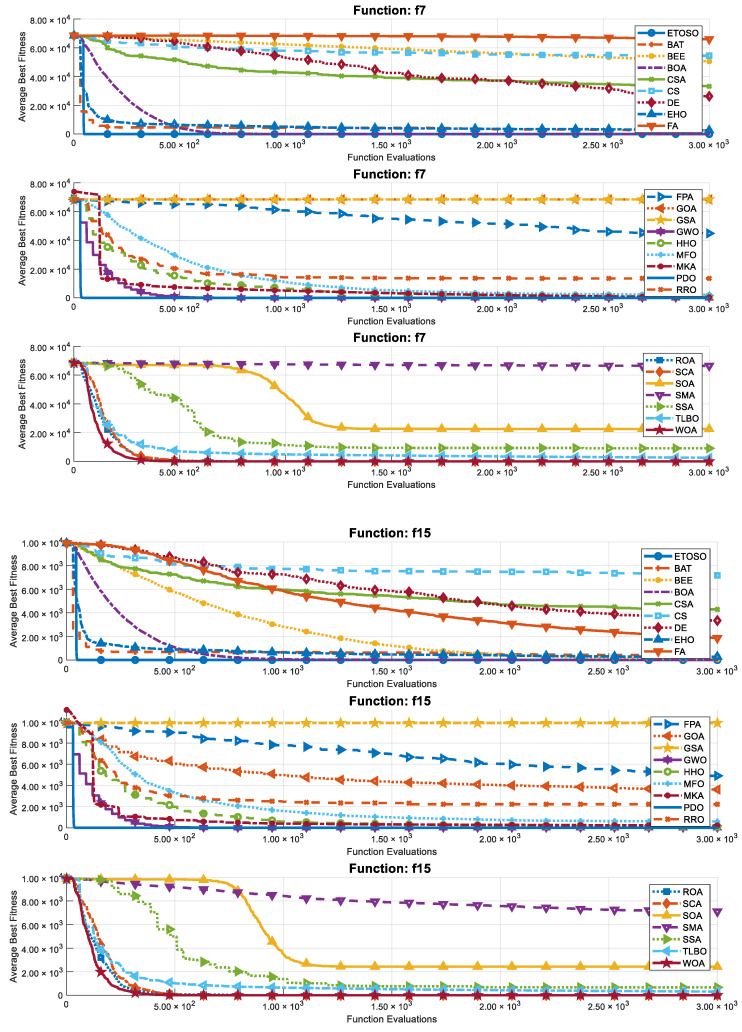
Convergence plots for D = 30 for f7 (multimodal) and f15 (rotated) averaged over 25 replications.

**Table 1 biomimetics-10-00222-t001:** Benchmark functions (D is the dimension).

Test Function			Range	Opt. *x*	f_min_
Unimodal	f1: Sphere	$f_{1}(x)=\sum_{i=1}^{D}x_{i}^{2}$	[−100, 100]	{0} ^D^	0
	f2: Zakharov	$f_{2}(x)=\sum_{i=1}^{D}x_{i}^{2}+{\left(\sum_{i=1}^{D}0.5ix_{i}\right)}^{2}+{\left(\sum_{i=1}^{D}0.5ix_{i}\right)}^{4}$	[−5, 10]	{0} ^D^	0
	f3: Sum squares	$f_{3}(x)=\sum_{i=1}^{D}ix_{i}^{2}$	[−10, 10]	{0} ^D^	0
Multimodal	f4: Schwefel’s Problem 2.22	$f_{4}\left(x\right)=\sum_{i=1}^{D}\left|x_{i}\right|+\prod_{i=1}^{D}\left|x_{i}\right|$	[−10, 10]	{0} ^D^	0
	f5: Schwefel’s Problem 2.26	$f_{5}\left(x\right)=-\sum_{i=1}^{D}\left(x_{i}sin\left(\sqrt{\left|x_{i}\right|}\right)\right)$	[−500, 500]	{420.96} ^D^	−418.9829 * ^D^
	f6: Rosenbrock	$f_{6}\left(x\right)=\sum_{i=1}^{D-1}\left[100{\left(x_{i+1}-x_{i}^{2}\right)}^{2}+{\left(x_{i}-1\right)}^{2}\right]$	[−30, 30]	{1} ^D^	0
	f7: Ackley	$f_{7}\left(x\right)=-20exp\left(-0.2\sqrt{\frac{1}{D}\sum_{i=1}^{D}x_{i}^{2}}\right)-exp\left(\frac{1}{D}\sum_{i=1}^{D}\cos2\pi x_{i}\right)+20+e$	[−32, 32]	{0} ^D^	0
	f8: Rastrigin	$f_{8}\left(x\right)=\sum_{i=1}^{D}\left[x_{i}^{2}-10\cos2\pi x_{i}+10\right]$	[−5.12, 5.12]	{0} ^D^	0
	f9: Griewank	$f_{9}\left(x\right)=\frac{1}{4000}\sum_{i=1}^{D}x_{i}^{2}-\prod_{i=1}^{D}cos\left(\frac{x_{i}}{\sqrt{i}}\right)+1$	[−600, 600]	{0} ^D^	0
	f10: Bent cigar	$f_{10}\left(x\right)=x_{1}^{2}+10^{6}\sum_{i=2}^{D}\left[x_{i}^{2}\right]$	[−100, 100]	{0} ^D^	0
Shifted ^a^	f11: Shifted Ackley	$f_{11}\left(x\right)=-20exp\left(-0.2\sqrt{\frac{1}{D}\sum_{i=1}^{D}z_{i}^{2}}\right)-exp\left(\frac{1}{D}\sum_{i=1}^{D}\cos2\pi z_{i}\right)+20+e+{fb}_{1}$	[−32, 32]	*o*	−140
	f12: Shifted Rosenbrock	$f_{12}\left(x\right)=\sum_{i=1}^{D-1}\left[100{\left(z_{i+1}-z_{i}^{2}\right)}^{2}+{\left(z_{i}-1\right)}^{2}\right]+{fb}_{2}$	[−100, 100]	*o*	390
	f13: Shifted Rastrigin	$f_{13}\left(x\right)=\sum_{i=1}^{D}\left[z_{i}^{2}-10\cos2\pi z_{i}+10\right]+{fb}_{3}$	[−5, 5]	*o*	−330
	f14: Shifted Griewank	$f_{14}\left(x\right)=\frac{1}{4000}\sum_{i=1}^{D}z_{i}^{2}-\prod_{i=1}^{D}cos\left(\frac{z_{i}}{\sqrt{i}}\right)+1+{fb}_{4}$	[−600, 600]	*o*	−180
Rotated ^b^	f15: Rotated Griewank	$f_{15}\left(x\right)=\frac{1}{4000}\sum_{i=1}^{D}y_{i}^{2}-\prod_{i=1}^{D}cos\left(\frac{y_{i}}{\sqrt{i}}\right)+1$	[−600, 600]	{0} ^D^	0

^a^ *z = x − o*, *o* is a shift added to global optimum and *fb_i_* is a bias added to the original function [58]. ^b^ *y* = *M*x*, where *M* is an orthogonal rotation matrix [59].

**Table 2 biomimetics-10-00222-t002:** Optimal parameter settings for comparative algorithms.

Algorithm		Parameters
Bat Algorithm	BAT [5,6]	A = 0.95, r_0_ = 0.9, Q_min_ = 0, Q_max_ = 5, α = 0.99, γ = 0.9, strategies = 8.
Bees Algorithm	BEE [7]	m = 3, e = 1, nep = 7, n_sp_ = 2, n_gh_ = 3.
Butterfly	BOA [12]	c = 0.01, a = 0.05, *p* = 0.5.
Crow Search	CSA [22]	A_P_ = 0.1, F_L_ = 2.
Cuckoo Search	CS [11]	p_a_ = 0.25, β = 1.
Differential Evolution	DE [35]	CR = 0.9, F = 0.5.
Elephant Herding	EHO [13]	α = rand (0 to 1), β = rand (0 to 1).
Firefly	FA [14,15]	α = 1, γ = 1, β_0_ = 0.2.
Flow Direction	FDA [25]	α = 25, β = 3.
Flower Pollination	FPA [8]	*p* = 0.8, λ = 1.5.
Grasshopper	GOA [16]	C = 1 to 0.00001 decreasing linearly.
Gravity Search	GSA [26]	G_0_ = 100, α = 20.
Grey Wolf Optimizer	GWO [23]	a = 2 to 0 linearly, c = 1 − 0.00009 t/T, t = current iteration, T = max iteration.
Harris Hawks	HHO [17]	E_0_ = rand (−1 to 1), E = 2 * E_0_ * (1 − t/T), r = rand (0 to 1).
Moth–Flame	MFO [18]	b = −1, t = rand (−1 to 1).
Monkey King	MKA [29,30,31,32]	Population = 10, step length = 1, climb number = 10, eyesight = 0.1, somersault interval = [−1, 1].
Prairie Dog	PDO [36]	ρ = 0.1, δ = 0.005.
Remora	ROA [33,34]	A_0_ = 0.9, A_f_ = 0.1, P_s_ = 0.499.
Raven Roosting	RRO [27,28]	R = search radius, D = dimension, N_steps_ = 5, P_follow_ = 0.2, P_stop_ = 0.1, R_pcpt_ = 3.6 * R * sqrt(D), R_leader_ = 1.8 * R * sqrt(D).
Sine Cosine	SCA [9]	t = current iteration, T max iteration, a = 2, r_1_ = a(1 − t)/T; r_2_ = rand (0 to 2π), r_3_ = rand (0 to 2), r_4_ = rand (0 to 1), *p* = 0.5.
Seagull Optimizer	SOA [37]	f_c_ = 2, u = 1, v = 1.
Slime Mold	SMA [19]	va_max_ = 1, va_min_ = 0.01, va = va_max_ − (va_max_ − va_min_)t/T.
Salp Swarm	SSA [20]	C = 2e^−(4t/T)^2.^
Teaching–Learning	TLBO [24]	No algorithm-specific parameters.
Whale Optimizer	WOA [10]	a = 2 to 0 linearly decreasing l = rand (−1 to 1), A = a(2l − 1), C = rand (0 to 2), b = 1.

**Table 3 biomimetics-10-00222-t003:** ETOSO versus TOSO for f1 to f10 with D = 2, 5, and 10 (5000D FEs and 25 replications).

Function	Metric	D = 2		D = 5		D = 10	
		TOSO	ETOSO	TOSO	ETOSO	TOSO	ETOSO
f1	Opt.	0	0	0	0	0	0
	Mean	0	0	0	0	0	0
	Std.	0	0	0	0	0	0
	Speed	1.10 × 10⁻^2^	1.08 × 10⁻^2^	2.89 × 10⁻^2^	2.73 × 10⁻^2^	1.82 × 10⁻^1^	5.81 × 10⁻^2^
f2	Opt.	0	0	0	0	0	0
	Mean	0	0	0	0	0	0
	Std.	0	0	0	0	0	0
	Speed	1.21 × 10⁻^2^	1.15 × 10⁻^2^	3.65 × 10⁻^2^	1.00 × 10⁻^1^	6.58 × 10⁻^2^	6.14 × 10⁻^2^
f3	Opt.	0	0	0	0	0	0
	Mean	0	0	0	0	0	0
	Std.	0	0	0	0	0	0
	Speed	1.13 × 10⁻^2^	1.06 × 10⁻^2^	8.09 × 10⁻^2^	8.48 × 10⁻^2^	6.14 × 10⁻^2^	6.07 × 10⁻^2^
f4	Opt.	0	0	0	0	0	0
	Mean	0	0	0	0	0	0
	Std.	0	0	0	0	0	0
	Speed	1.11 × 10⁻^2^	1.08 × 10⁻^2^	7.63 × 10⁻^2^	1.02 × 10⁻^1^	5.91 × 10⁻^2^	5.75 × 10⁻^2^
f5	Opt.	−8.38 × 10^2^	−8.38 × 10^2^	−2.09 × 10^3^	−2.09 × 10^3^	−4.19 × 10^3^	−4.19 × 10^3^
	Mean	−8.38 × 10^2^	−8.38 × 10^2^	−2.09 × 10^3^	−2.09 × 10^3^	−4.19 × 10^3^	−4.19 × 10^3^
	Std.	3.78 × 10⁻^5^	2.09 × 10⁻^5^	1.11 × 10⁻^4^	8.08 × 10⁻^5^	8.93 × 10⁻^5^	1.41 × 10⁻^4^
	Speed	1.14 × 10⁻^2^	1.13 × 10⁻^2^	1.10 × 10⁻^1^	8.46 × 10⁻^2^	8.02 × 10⁻^2^	7.64 × 10⁻^2^
f6	Opt.	0	0	0	0	0	0
	Mean	3.35 × 10⁻^1^	3.43 × 10⁻^1^	2.38 × 10^0^	2.41 × 10^0^	7.33 × 10^0^	7.38 × 10^0^
	Std.	3.06 × 10⁻^2^	2.88 × 10⁻^2^	1.47 × 10⁻^1^	1.26 × 10⁻^1^	1.38 × 10⁻^1^	1.66 × 10⁻^1^
	Speed	1.18 × 10⁻^2^	1.08 × 10⁻^2^	1.01 × 10⁻^1^	1.03 × 10⁻^1^	6.18 × 10⁻^2^	5.81 × 10⁻^2^
f7	Opt.	0	0	0	0	0	0
	Mean	0	0	0	0	0	0
	Std.	0	0	0	0	0	0
	Speed	1.14 × 10⁻^2^	1.13 × 10⁻^2^	1.06 × 10⁻^1^	1.10 × 10⁻^1^	8.02 × 10⁻^2^	7.64 × 10⁻^2^
f8	Opt.	0	0	0	0	0	0
	Mean	0	0	0	0	0	0
	Std.	0	0	0	0	0	0
	Speed	1.12 × 10⁻^2^	1.13 × 10⁻^2^	9.45 × 10⁻^2^	9.47 × 10⁻^2^	7.72 × 10⁻^2^	7.41 × 10⁻^2^
f9	Opt.	0	0	0	0	0	0
	Mean	0	0	0	0	0	0
	Std.	0	0	0	0	0	0
	Speed	1.27 × 10⁻^2^	1.20 × 10⁻^2^	7.91 × 10⁻^2^	1.03 × 10⁻^1^	6.54 × 10⁻^2^	5.92 × 10⁻^2^
f10	Opt.	0	0	0	0	0	0
	Mean	0	0	0	0	0	0
	Std.	0	0	0	0	0	0
	Speed	1.21 × 10⁻^2^	1.18 × 10⁻^2^	1.10 × 10⁻^1^	9.19 × 10⁻^2^	5.80 × 10⁻^2^	5.66 × 10⁻^2^

**Table 4 biomimetics-10-00222-t004:** ETOSO versus TOSO for f11 to f15 with D = 30, 50, 100, and 200 (5000D FEs and 25 replications).

Function	Metric	D = 30		D = 50		D = 100		D = 200	
		TOSO	ETOSO	TOSO	ETOSO	TOSO	ETOSO	TOSO	ETOSO
f11	Opt.	−140.00	−140.00	−140.00	−140.00	−140.00	−140.00	−140.00	−140.00
	Mean	−140.00	−140.00	−140.00	−140.00	−140.00	−140.00	−140.00	−140.00
	Std.	0.00	0.00	0.00	0.00	0.00	0.00	0.00	0.00
	Speed	0.31	0.30	0.57	0.55	1.37	1.32	4.09	3.74
f12	Opt.	390.00	390.00	390.00	390.00	390.00	390.00	390.00	390.00
	Mean	418.24	418.24	438.05	438.05	487.57	487.58	586.63	586.65
	Std.	0.00	0.01	0.01	0.00	0.01	0.01	0.01	0.01
	Speed	0.21	0.21	0.40	0.38	1.01	0.96	2.85	2.71
f13	Opt.	−330.00	−330.00	−330.00	−330.00	−330.00	−330.00	−330.00	−330.00
	Mean	−330.00	−330.00	−330.00	−330.00	−330.00	−330.00	−330.00	−330.00
	Std.	0.00	0.00	0.00	0.00	0.00	0.00	0.00	0.00
	Speed	0.31	0.29	0.56	0.54	1.32	1.28	3.79	3.38
f14	Opt.	−180.00	−180.00	−180.00	−180.00	−180.00	−180.00	−180.00	−180.00
	Mean	−179.98	−179.96	−179.98	−179.98	−179.99	−179.98	−179.99	−179.98
	Std.	0.02	0.04	0.03	0.03	0.01	0.02	0.03	0.02
	Speed	0.27	0.26	0.55	0.52	1.57	1.45	4.96	4.57
f15	Opt.	0.00	0.00	0.00	0.00	0.00	0.00	0.00	0.00
	Mean	0.00	0.00	0.00	0.00	0.00	0.00	0.00	0.00
	Std.	0.00	0.00	0.00	0.00	0.00	0.00	0.00	0.00
	Speed	0.28	0.26	0.66	0.61	1.89	1.72	6.33	5.79

**Table 5 biomimetics-10-00222-t005:** Benchmark results for functions f1–f8 with D = 5 (using 5000D functions evaluations and 25 replications).

Algorithm		f1	f2	f3	f4	f5	f6	f7	f8
	Opt.	0	0	0	0	−2.09 × 10^3^	0	0	0
ETOSO	Best	0	0	0	0	−2.09 × 10^3^	2.11	0	0
	Mean	0	0	0	0	−2.09 × 10^3^	2.41	0	0
BAT	Best	9.98 × 10⁻^11^	4.73 × 10⁻^10^	7.71 × 10⁻^10^	1.27 × 10⁻^5^	−1.98 × 10^3^	3.47 × 10⁻^2^	2.38 × 10⁻^5^	2.98
	Mean	8.55 × 10⁻^10^	3.30 × 10⁻^9^	5.01 × 10⁻^9^	6.64 × 10⁻^5^	−1.56 × 10^3^	1.17 × 10^2^	2.51 × 10⁻^1^	1.03 × 10^1^
BEE	Best	3.00 × 10⁻^4^	2.00 × 10⁻^4^	3.00 × 10⁻^4^	2.65 × 10⁻^2^	−1.98 × 10^3^	1.16 × 10⁻^1^	1.38 × 10^1^	3.22
	Mean	6.00 × 10⁻^4^	8.00 × 10⁻^4^	1.30 × 10⁻^3^	4.23 × 10⁻^2^	−1.58 × 10^3^	4.66 × 10⁻^1^	1.71 × 10^1^	1.35 × 10^1^
BOA	Best	6.50 × 10⁻^3^	1.70 × 10⁻^5^	3.17 × 10⁻^5^	1.13 × 10⁻^2^	−1.57 × 10^3^	1.92 × 10⁻^1^	1.39 × 10^1^	3.00
	Mean	1.92 × 10⁻^2^	5.71 × 10⁻^5^	2.00 × 10⁻^4^	2.39 × 10⁻^2^	−1.13 × 10^3^	1.62 × 10^1^	1.71 × 10^1^	2.13 × 10^1^
CSA	Best	1.32 × 10^1^	1.04 × 10⁻^1^	3.40 × 10⁻^1^	6.11 × 10⁻^1^	−2.02 × 10^3^	4.49 × 10^1^	3.78	2.83
	Mean	4.70 × 10^1^	6.38 × 10⁻^1^	1.06	1.04	−1.73 × 10^3^	8.27 × 10^2^	5.01	7.35
CS	Best	2.58 × 10^2^	1.26	5.11	2.68	−1.73 × 10^3^	2.51 × 10^3^	8.51	1.02 × 10^1^
	Mean	4.99 × 10^2^	6.19	1.25 × 10^1^	4.10	−1.58 × 10^3^	2.28 × 10^4^	1.07 × 10^1^	1.61 × 10^1^
DE	Best	2.94 × 10⁻^42^	1.97 × 10⁻^36^	1.05 × 10⁻^43^	1.49 × 10⁻^21^	−2.09 × 10^3^	1.40 × 10⁻^23^	0	0
	Mean	2.15 × 10⁻^39^	5.54 × 10⁻^34^	3.06 × 10⁻^41^	3.45 × 10⁻^20^	−1.98 × 10^3^	3.51 × 10⁻^21^	1.13 × 10⁻^15^	2.00 × 10⁻^1^
EHO	Best	6.18 × 10⁻^9^	5.00 × 10⁻^4^	8.55 × 10⁻^9^	1.10 × 10⁻^3^	−2.08 × 10^3^	8.70 × 10⁻^3^	1.79 × 10⁻^6^	5.45 × 10⁻^6^
	Mean	2.87	6.13 × 10⁻^1^	1.95 × 10⁻^1^	2.71 × 10⁻^1^	−1.64 × 10^3^	1.40 × 10^2^	1.43	6.36
FA	Best	7.40 × 10⁻^3^	1.17 × 10⁻^2^	1.89 × 10⁻^2^	1.62 × 10⁻^1^	−1.74 × 10^3^	1.20	1.49 × 10^1^	2.95
	Mean	2.88 × 10⁻^2^	3.75 × 10⁻^2^	6.11 × 10⁻^2^	2.87 × 10⁻^1^	−1.59 × 10^3^	3.70	1.76 × 10^1^	6.62
FDA	Best	3.90 × 10⁻^4^	5.10 × 10⁻^4^	4.20 × 10⁻^4^	2.88 × 10⁻^2^	−1.28 × 10^5^	1.06 × 10⁻^1^	1.97 × 10⁻^2^	1.04
	Mean	1.43 × 10⁻^3^	2.21 × 10⁻^3^	3.49 × 10⁻^3^	6.20 × 10⁻^2^	−8.49 × 10^4^	2.34 × 10^1^	6.09 × 10⁻^2^	5.38
FPA	Best	5.32 × 10⁻^42^	5.85 × 10⁻^26^	5.82 × 10⁻^43^	2.49 × 10⁻^26^	−2.09 × 10^3^	5.03 × 10⁻^4^	0	0
	Mean	8.24 × 10⁻^38^	2.17 × 10⁻^23^	1.20 × 10⁻^39^	8.78 × 10⁻^25^	−1.99 × 10^3^	6.45 × 10⁻^1^	2.13 × 10⁻^15^	3.58 × 10⁻^1^
GOA	Best	2.10 × 10^3^	8.89 × 10⁻^11^	3.42 × 10⁻^13^	1.79 × 10⁻^6^	−1.35 × 10^3^	4.00 × 10⁻^4^	1.40 × 10⁻^6^	9.95 × 10⁻^1^
	Mean	5.54 × 10^3^	4.75 × 10⁻^10^	4.37 × 10⁻^11^	5.07 × 10⁻^5^	−9.41 × 10^2^	3.51 × 10^1^	1.39 × 10⁻^1^	7.72
GSA	Best	2.10 × 10^3^	4.54 × 10⁻^8^	4.84 × 10⁻^7^	9.00 × 10⁻^4^	−1.35 × 10^3^	1.71 × 10^4^	1.61 × 10^1^	3.70 × 10⁻^4^
	Mean	5.56 × 10^3^	1.90 × 10⁻^6^	2.20 × 10⁻^6^	1.97 × 10⁻^3^	−9.41 × 10^2^	4.29 × 10^6^	1.86 × 10^1^	3.08
GWO	Best	1.10 × 10⁻^192^	2.90 × 10⁻^144^	4.90 × 10⁻^198^	9.70 × 10⁻^107^	−2.09 × 10^3^	4.37 × 10⁻^3^	0	0
	Mean	1.00 × 10⁻^177^	1.90 × 10⁻^131^	6.10 × 10⁻^184^	2.30 × 10⁻^99^	−1.73 × 10^3^	1.53	1.56 × 10⁻^15^	0
HHO	Best	1.02 × 10⁻^8^	3.56 × 10⁻^7^	5.09 × 10⁻^9^	1.10 × 10⁻^4^	−2.09 × 10^3^	5.14 × 10⁻^2^	4.50 × 10⁻^5^	3.72 × 10⁻^7^
	Mean	3.20 × 10⁻^6^	1.71 × 10⁻^3^	5.62 × 10⁻^7^	1.44 × 10⁻^3^	−1.91 × 10^3^	9.45 × 10⁻^1^	6.67 × 10⁻^2^	3.42
MFO	Best	3.40 × 10⁻^223^	1.30 × 10⁻^159^	1.70 × 10⁻^221^	9.60 × 10⁻^118^	−2.09 × 10^3^	1.55 × 10⁻^3^	0	0
	Mean	4.70 × 10⁻^211^	1.38	8.40 × 10⁻^213^	6.10 × 10⁻^114^	−1.72 × 10^3^	1.34 × 10^1^	6.58 × 10⁻^2^	2.79
MKA	Best	1.70 × 10⁻^4^	4.47 × 10⁻^5^	5.00 × 10⁻^4^	1.96 × 10⁻^2^	−1.98 × 10^3^	1.09 × 10⁻^1^	2.59 × 10⁻^2^	1.27 × 10⁻^1^
	Mean	6.30 × 10⁻^4^	9.00 × 10⁻^4^	1.27 × 10⁻^3^	4.01 × 10⁻^2^	−1.58 × 10^3^	1.33	4.42 × 10⁻^2^	2.14
PDO	Best	0	2.87	0	0	−1.49 × 10^3^	4.40 × 10⁻^2^	0	0
	Mean	0	1.38 × 10^2^	0	0	−9.23 × 10^2^	3.83 × 10⁻^1^	0	0
RRO	Best	2.38 × 10⁻^1^	2.47 × 10⁻^3^	3.28 × 10⁻^2^	7.97 × 10⁻^2^	−2.09 × 10^3^	2.93 × 10⁻^1^	1.72	8.29 × 10⁻^2^
	Mean	5.71 × 10^1^	7.11 × 10⁻^1^	1.40	1.35	−2.09 × 10^3^	6.48 × 10^2^	5.11	6.46
ROA	Best	0	7.60 × 10⁻^273^	3.60 × 10⁻^303^	7.30 × 10⁻^162^	−2.09 × 10^3^	7.90 × 10⁻^6^	0	0
	Mean	3.80 × 10⁻^215^	3.30 × 10⁻^224^	3.50 × 10⁻^221^	3.10 × 10⁻^117^	−2.09 × 10^3^	1.23	0	0
SCA	Best	5.50 × 10⁻^183^	1.30 × 10⁻^192^	5.80 × 10⁻^204^	6.20 × 10⁻^108^	−2.09 × 10^3^	1.60	0	0
	Mean	3.55 × 10⁻^76^	9.39 × 10⁻^1^	3.86 × 10⁻^82^	3.30 × 10⁻^82^	−2.09 × 10^3^	3.09	0	0
SOA	Best	9.96 × 10⁻^1^	4.20	1.99	3.03	−1.96 × 10^3^	4.35 × 10^2^	3.84	1.46 × 10^1^
	Mean	1.29 × 10^2^	4.41 × 10^1^	4.40 × 10^1^	1.09 × 10^1^	−1.44 × 10^3^	2.98 × 10^5^	1.27 × 10^1^	3.58 × 10^1^
SMA	Best	1.17	4.10 × 10⁻^4^	4.80 × 10⁻^4^	2.94 × 10⁻^2^	−1.82 × 10^3^	2.00	1.38 × 10^1^	4.20
	Mean	1.61 × 10^3^	1.85 × 10⁻^3^	2.78 × 10⁻^3^	5.83 × 10⁻^2^	−1.40 × 10^3^	1.97 × 10^3^	1.72 × 10^1^	1.93 × 10^1^
SSA	Best	3.93 × 10⁻^6^	1.48 × 10⁻^1^	1.37 × 10⁻^2^	5.01 × 10⁻^2^	−1.39 × 10^3^	4.33 × 10^1^	1.24	1.60 × 10⁻^4^
	Mean	3.61 × 10^2^	5.77	5.56	2.72	−1.13 × 10^3^	1.85 × 10^4^	8.46	9.04
TLBO	Best	9.11 × 10⁻^38^	1.14 × 10⁻^26^	1.77 × 10⁻^37^	2.77 × 10⁻^23^	−2.09 × 10^3^	1.40 × 10⁻^4^	3.55 × 10⁻^15^	0
	Mean	1.60 × 10⁻^28^	2.45 × 10⁻^20^	1.47 × 10⁻^29^	6.06 × 10⁻^21^	−1.95 × 10^3^	1.77	6.58 × 10⁻^2^	1.13
WOA	Best	8.00 × 10⁻^301^	7.80 × 10⁻^115^	1.90 × 10⁻^299^	1.40 × 10⁻^161^	−2.09 × 10^3^	3.14 × 10⁻^2^	0	0
	Mean	3.20 × 10⁻^284^	2.90 × 10⁻^66^	2.00 × 10⁻^282^	2.00 × 10⁻^154^	−1.74 × 10^3^	2.54	2.70 × 10⁻^15^	7.10 × 10⁻^17^

**Table 6 biomimetics-10-00222-t006:** Benchmark results for functions f9–f15 with D = 5 (using 5000D functions evaluations and 25 replications).

Alg.		f9	f10	f11	f12	f13	f14	f15
	Opt.	0	0	−1.40 × 10^2^	3.90 × 10^2^	−3.30 × 10^2^	−1.80 × 10^2^	0
ETOSO	Best	0	0	−1.40 × 10^2^	3.93 × 10^2^	−3.30 × 10^2^	−1.80 × 10^2^	0
	Mean	0	0	−1.40 × 10^2^	3.93 × 10^2^	−3.30 × 10^2^	−1.80 × 10^2^	0
BAT	Best	9.11 × 10⁻^2^	4.41	−1.40 × 10^2^	3.93 × 10^2^	−3.28 × 10^2^	−1.80 × 10^2^	2.83 × 10⁻^1^
	Mean	5.68 × 10⁻^1^	1.47 × 10^3^	−1.40 × 10^2^	7.13 × 10^2^	−3.19 × 10^2^	−1.79 × 10^2^	9.30 × 10⁻^1^
BEE	Best	1.21 × 10^1^	1.58 × 10^2^	−1.27 × 10^2^	3.93 × 10^2^	−3.28 × 10^2^	−1.62 × 10^2^	5.15 × 10^1^
	Mean	4.66 × 10^1^	6.77 × 10^2^	−1.22 × 10^2^	3.94 × 10^2^	−3.10 × 10^2^	−1.18 × 10^2^	1.10 × 10^2^
BOA	Best	8.27 × 10⁻^2^	1.58 × 10^4^	−1.25 × 10^2^	3.94 × 10^2^	−3.03 × 10^2^	−1.67 × 10^2^	8.84 × 10⁻^2^
	Mean	2.88 × 10⁻^1^	1.72 × 10^5^	−1.21 × 10^2^	3.99 × 10^2^	−2.65 × 10^2^	−1.26 × 10^2^	2.29 × 10⁻^1^
CSA	Best	8.55 × 10⁻^1^	1.29 × 10^6^	−1.37 × 10^2^	1.28 × 10^3^	−3.26 × 10^2^	−1.79 × 10^2^	1.27
	Mean	1.28	9.41 × 10^6^	−1.35 × 10^2^	2.91 × 10^3^	−3.22 × 10^2^	−1.79 × 10^2^	2.14
CS	Best	2.54	2.42 × 10^7^	−1.33 × 10^2^	5.57 × 10^3^	−3.22 × 10^2^	−1.77 × 10^2^	2.32
	Mean	5.18	1.27 × 10^8^	−1.29 × 10^2^	2.27 × 10^4^	−3.12 × 10^2^	−1.76 × 10^2^	1.11 × 10^1^
DE	Best	1.19 × 10⁻^2^	1.63 × 10⁻^38^	−1.40 × 10^2^	3.93 × 10^2^	−3.30 × 10^2^	−1.80 × 10^2^	3.94 × 10⁻^2^
	Mean	8.56 × 10⁻^2^	4.60 × 10⁻^35^	−1.40 × 10^2^	3.93 × 10^2^	−3.30 × 10^2^	−1.80 × 10^2^	1.95 × 10⁻^1^
EHO	Best	1.76 × 10⁻^1^	3.95	−1.40 × 10^2^	6.46 × 10^2^	−3.25 × 10^2^	−1.80 × 10^2^	8.20 × 10⁻^3^
	Mean	5.21 × 10⁻^1^	6.89 × 10^4^	−1.35 × 10^2^	4.32 × 10^3^	−3.18 × 10^2^	−1.79 × 10^2^	8.33 × 10⁻^1^
FA	Best	1.31 × 10^1^	6.05 × 10^3^	−1.25 × 10^2^	3.95 × 10^2^	−3.27 × 10^2^	−1.60 × 10^2^	5.20 × 10^1^
	Mean	4.72 × 10^1^	1.42 × 10^4^	−1.22 × 10^2^	4.14 × 10^2^	−3.24 × 10^2^	−1.16 × 10^2^	9.65 × 10^1^
FDA	Best	1.76 × 10⁻^2^	2.05 × 10^2^	−1.40 × 10^2^	3.93 × 10^2^	−3.30 × 10^2^	−1.80 × 10^2^	3.56 × 10⁻^2^
	Mean	1.17 × 10⁻^1^	6.08 × 10^3^	−1.39 × 10^2^	3.94 × 10^2^	−3.26 × 10^2^	−1.80 × 10^2^	1.38 × 10⁻^1^
FPA	Best	3.70 × 10⁻^9^	1.50 × 10⁻^36^	−1.40 × 10^2^	3.93 × 10^2^	−3.30 × 10^2^	−1.80 × 10^2^	1.74 × 10⁻^2^
	Mean	4.69 × 10⁻^2^	4.65 × 10⁻^34^	−1.40 × 10^2^	3.93 × 10^2^	−3.29 × 10^2^	−1.80 × 10^2^	1.26 × 10⁻^1^
GOA	Best	1.31 × 10^1^	2.78 × 10^8^	−1.40 × 10^2^	1.37 × 10^5^	−3.29 × 10^2^	−1.60 × 10^2^	5.20 × 10^1^
	Mean	4.81 × 10^1^	2.83 × 10^9^	−1.40 × 10^2^	3.11 × 10^5^	−3.24 × 10^2^	−1.16 × 10^2^	1.29 × 10^2^
GSA	Best	1.31 × 10^1^	2.78 × 10^8^	−1.25 × 10^2^	1.37 × 10^5^	−3.29 × 10^2^	−1.60 × 10^2^	5.20 × 10^1^
	Mean	4.81 × 10^1^	2.88 × 10^8^	−1.21 × 10^2^	3.11 × 10^5^	−3.25 × 10^2^	−1.16 × 10^2^	1.29 × 10^2^
GWO	Best	0	4.73 × 10⁻^188^	−1.40 × 10^2^	3.93 × 10^2^	−3.29 × 10^2^	−1.80 × 10^2^	9.38 × 10⁻^3^
	Mean	1.29 × 10⁻^2^	3.36 × 10⁻^176^	−1.39 × 10^2^	1.28 × 10^3^	−3.27 × 10^2^	−1.80 × 10^2^	6.32 × 10⁻^2^
HHO	Best	8.37 × 10⁻^6^	1.81 × 10⁻^2^	−1.40 × 10^2^	3.94 × 10^2^	−3.29 × 10^2^	−1.80 × 10^2^	4.82 × 10⁻^2^
	Mean	2.07 × 10⁻^1^	3.84 × 10^3^	−1.39 × 10^2^	4.40 × 10^2^	−3.22 × 10^2^	−1.80 × 10^2^	3.70 × 10⁻^1^
MFO	Best	7.40 × 10⁻^3^	6.90 × 10⁻^214^	−1.40 × 10^2^	3.93 × 10^2^	−3.30 × 10^2^	−1.80 × 10^2^	4.68 × 10⁻^2^
	Mean	6.92 × 10⁻^2^	3.60 × 10^3^	−1.38 × 10^2^	5.33 × 10^2^	−3.25 × 10^2^	−1.80 × 10^2^	2.48 × 10⁻^1^
MKA	Best	2.84 × 10⁻^1^	7.30 × 10^1^	−1.40 × 10^2^	3.93 × 10^2^	−3.29 × 10^2^	−1.79 × 10^2^	1.43 × 10⁻^1^
	Mean	5.60	6.54 × 10^2^	−1.26 × 10^2^	3.94 × 10^2^	−3.27 × 10^2^	−1.71 × 10^2^	4.08
PDO	Best	0	0	−1.25 × 10^2^	1.52 × 10^5^	−2.98 × 10^2^	−1.61 × 10^2^	0
	Mean	0	0	−1.23 × 10^2^	2.44 × 10^5^	−2.78 × 10^2^	−1.39 × 10^2^	0
RRO	Best	9.95 × 10⁻^1^	2.55 × 10^5^	−1.36 × 10^2^	2.03 × 10^3^	−3.24 × 10^2^	−1.79 × 10^2^	9.87 × 10⁻^1^
	Mean	1.58	2.14 × 10^7^	−1.34 × 10^2^	6.43 × 10^3^	−3.20 × 10^2^	−1.78 × 10^2^	2.37
ROA	Best	0	1.38 × 10⁻^282^	−1.37 × 10^2^	5.96 × 10^2^	−3.26 × 10^2^	−1.80 × 10^2^	0
	Mean	0	5.94 × 10⁻^209^	−1.30 × 10^2^	1.67 × 10^4^	−3.14 × 10^2^	−1.77 × 10^2^	7.81 × 10⁻^2^
SCA	Best	0	1.17 × 10⁻^182^	−1.39 × 10^2^	4.53 × 10^2^	−3.29 × 10^2^	−1.80 × 10^2^	0
	Mean	0	1.05 × 10⁻^75^	−1.37 × 10^2^	1.22 × 10^3^	−3.23 × 10^2^	−1.79 × 10^2^	4.44 × 10⁻^18^
SOA	Best	3.54 × 10⁻^1^	1.34 × 10^2^	−1.33 × 10^2^	4.03 × 10^2^	−3.18 × 10^2^	−1.79 × 10^2^	4.73 × 10⁻^1^
	Mean	1.58	3.99 × 10^6^	−1.26 × 10^2^	2.21 × 10^4^	−2.95 × 10^2^	−1.72 × 10^2^	1.17 × 10^1^
SMA	Best	1.21 × 10^1^	1.73 × 10^3^	−1.27 × 10^2^	3.45 × 10^3^	−3.28 × 10^2^	−1.62 × 10^2^	5.15 × 10^1^
	Mean	4.65 × 10^1^	5.79 × 10^8^	−1.22 × 10^2^	7.08 × 10^4^	−3.09 × 10^2^	−1.17 × 10^2^	1.08 × 10^2^
SSA	Best	1.21 × 10⁻^1^	1.63 × 10^4^	−1.37 × 10^2^	1.95 × 10^3^	−3.22 × 10^2^	−1.79 × 10^2^	2.58 × 10⁻^1^
	Mean	3.73	1.02 × 10^8^	−1.30 × 10^2^	2.59 × 10^4^	−3.07 × 10^2^	−1.73 × 10^2^	1.23 × 10^1^
TLBO	Best	2.31 × 10⁻^2^	3.81 × 10⁻^30^	−1.40 × 10^2^	3.93 × 10^2^	−3.30 × 10^2^	−1.80 × 10^2^	2.96 × 10⁻^2^
	Mean	9.34 × 10⁻^2^	4.00 × 10^2^	−1.40 × 10^2^	3.93 × 10^2^	−3.29 × 10^2^	−1.80 × 10^2^	1.64 × 10⁻^1^
WOA	Best	0	1.14 × 10⁻^297^	−1.40 × 10^2^	5.27 × 10^2^	−3.29 × 10^2^	−1.80 × 10^2^	0
	Mean	1.81 × 10⁻^2^	1.05 × 10⁻^278^	−1.30 × 10^2^	1.23 × 10^4^	−3.11 × 10^2^	−1.78 × 10^2^	2.87 × 10⁻^1^

**Table 7 biomimetics-10-00222-t007:** Ranking of all algorithms on the basis of performance for D = 2, 5, and 10.

Algorithm	Rank for D = 2	Rank for D = 5	Rank for D = 10	Cumulative Rank	Overall Rank
ETOSO	1	1	1	3	1
FPA	2	2	2	6	2
DE	4	2	3	9	3
GWO	5	4	4	13	4
TLBO	3	6	6	15	5
ROA	9	5	5	19	6
SCA	6	7	7	20	7
WOA	8	8	8	24	8
MFO	7	9	11	27	9
HHO	10	11	9	30	10
PDO	11	10	11	32	11
MKA	13	13	10	36	12
FDA	12	12	13	37	13
BAT	14	13	14	41	14
EHO	16	15	15	46	15
CSA	17	17	19	53	16
RRO	19	18	16	53	17
BEE	21	16	17	54	18
BOA	18	19	20	57	19
FA	20	21	18	59	20
GOA	15	20	26	61	21
CS	22	24	23	69	22
SSA	25	23	21	69	23
GSA	24	22	24	70	24
SMA	26	25	22	73	25
SOA	23	26	25	74	26

**Table 8 biomimetics-10-00222-t008:** Top algorithms evaluation for f1 to f8 and D = 200 (25 replications and 5000D FEs).

Alg.		f1	f2	f3	f4	f5	f6	f7	f8
	Opt.	0	0	0	0	−8.38 × 10^4^	0	0	0
ETOSO	Best	0	0	0	0	−8.38 × 10^4^	1.95 × 10^2^	0	0
	Mean	0	0	0	0	−8.38 × 10^4^	1.95 × 10^2^	0	0
	Std	0	0	0	0	5.50 × 10⁻^4^	1.68 × 10⁻^1^	0	0
	Speed	2.27	2.51	2.36	2.35	3.84	2.44	3.43	3.41
DE	Best	1.00 × 10⁻^4^	1.58 × 10^3^	1.00 × 10⁻^4^	4.10 × 10⁻^4^	−6.02 × 10^4^	7.67 × 10^2^	1.99 × 10^1^	2.76 × 10^2^
	Mean	2.28 × 10^4^	2.42 × 10^3^	1.85 × 10⁻^2^	6.12 × 10⁻^2^	−5.53 × 10^4^	2.10 × 10^9^	1.99 × 10^1^	3.63 × 10^2^
	Std	1.13 × 10^5^	2.77 × 10^2^	3.76 × 10⁻^2^	1.70 × 10⁻^1^	2.36 × 10^3^	1.07 × 10^9^	4.00 × 10⁻^4^	5.29 × 10^1^
	Speed	1.35 × 10^1^	1.48 × 10^1^	1.42 × 10^1^	1.40 × 10^1^	1.55 × 10^1^	1.49 × 10^1^	1.54 × 10^1^	1.57 × 10^1^
FPA	Best	3.26 × 10^5^	1.96 × 10^3^	2.98 × 10^5^	3.06 × 10^2^	−4.45 × 10^4^	1.44 × 10^9^	1.99 × 10^1^	2.17 × 10^3^
	Mean	3.62 × 10^5^	3.08 × 10^3^	3.42 × 10^5^	2.40 × 10^52^	−4.07 × 10^4^	1.62 × 10^9^	1.99 × 10^1^	2.41 × 10^3^
	Std	1.45 × 10^4^	4.24 × 10^2^	1.81 × 10^4^	1.20 × 10^53^	1.82 × 10^3^	9.78 × 10^7^	1.32 × 10⁻^3^	1.01 × 10^2^
	Speed	3.02 × 10^1^	3.19 × 10^1^	3.11 × 10^1^	3.08 × 10^1^	3.34 × 10^1^	3.23 × 10^1^	3.41 × 10^1^	3.39 × 10^1^
GWO	Best	0	3.60 × 10⁻^176^	0	0	−3.08 × 10^4^	1.96 × 10^2^	7.10 × 10⁻^15^	0
	Mean	0	2.93 × 10⁻^158^	0	0	−2.53 × 10^4^	1.98 × 10^2^	7.10 × 10⁻^15^	0
	Std	0	1.19 × 10⁻^157^	0	0	2.75 × 10^3^	6.26 × 10⁻^1^	0	0
	Speed	2.06 × 10^1^	2.07 × 10^1^	2.11 × 10^1^	2.15 × 10^1^	2.55 × 10^1^	2.09 × 10^1^	2.40 × 10^1^	2.30 × 10^1^
HHO	Best	3.93 × 10⁻^6^	5.93 × 10^1^	1.87 × 10⁻^5^	5.54 × 10⁻^2^	−8.38 × 10^4^	1.97 × 10^2^	7.90 × 10⁻^4^	4.00 × 10⁻^4^
	Mean	3.20 × 10⁻^4^	6.17 × 10^2^	5.39 × 10⁻^3^	8.67 × 10⁻^1^	−7.13 × 10^4^	1.97 × 10^2^	2.05 × 10⁻^3^	8.24 × 10^1^
	Std	2.50 × 10⁻^4^	4.24 × 10^2^	8.06 × 10⁻^3^	2.15	1.22 × 10^4^	6.18 × 10⁻^3^	7.60 × 10⁻^4^	8.46 × 10^1^
	Speed	1.66	1.96	1.74	1.79	3.48	2.08	3.35	3.08
MFO	Best	3.00 × 10^4^	4.16 × 10^3^	3.40 × 10^4^	2.10 × 10^2^	−5.24 × 10^4^	7.45 × 10^1^	1.94 × 10^1^	1.60 × 10^3^
	Mean	6.48 × 10^4^	5.05 × 10^3^	9.05 × 10^4^	4.27 × 10^2^	−4.37 × 10^4^	6.98 × 10^8^	1.97 × 10^1^	1.84 × 10^3^
	Std	2.62 × 10^4^	4.65 × 10^2^	2.43 × 10^4^	9.77 × 10^1^	4.20 × 10^3^	1.10 × 10^9^	1.76 × 10⁻^1^	1.51 × 10^2^
	Speed	2.32 × 10^1^	2.44 × 10^1^	2.41 × 10^1^	2.35 × 10^1^	2.60 × 10^1^	2.55 × 10^1^	2.59 × 10^1^	2.58 × 10^1^
PDO	Best	0	4.52 × 10^8^	0	0	−4.60 × 10^4^	1.97 × 10^2^	0	0
	Mean	0	5.55 × 10^14^	0	0	−4.23 × 10^4^	1.97 × 10^2^	0	0
	Std	0	1.73 × 10^15^	0	0	2.90 × 10^3^	1.18 × 10⁻^2^	0	0
	Speed	1.52 × 10^1^	1.64 × 10^1^	1.60 × 10^1^	1.56 × 10^1^	1.89 × 10^1^	1.73 × 10^1^	1.73 × 10^1^	1.62 × 10^1^
ROA	Best	0	0	0	0	−8.38 × 10^4^	6.60 × 10⁻^7^	0	0
	Mean	0	0	0	0	−8.38 × 10^4^	3.30 × 10⁻^3^	0	0
	Std	0	0	0	0	4.27 × 10⁻^3^	5.30 × 10⁻^3^	0	0
	Speed	1.54 × 10^1^	1.60 × 10^1^	1.60 × 10^1^	1.48 × 10^1^	1.69 × 10^1^	1.66 × 10^1^	1.54 × 10^1^	1.54 × 10^1^
SCA	Best	0	6.57	0	0	−8.38 × 10^4^	1.97 × 10^2^	0	0
	Mean	0	1.42 × 10^2^	0	0	−8.38 × 10^4^	9.65 × 10^7^	0	0
	Std	0	1.40 × 10^2^	0	0	1.18	2.80 × 10^8^	0	0
	Speed	1.49 × 10^1^	1.52 × 10^1^	1.52 × 10^1^	1.46 × 10^1^	1.66 × 10^1^	1.63 × 10^1^	1.55 × 10^1^	1.51 × 10^1^
TLBO	Best	1.40 × 10⁻^3^	1.88 × 10^3^	1.37 × 10⁻^1^	5.40 × 10⁻^3^	−5.44 × 10^4^	1.55 × 10^3^	1.34 × 10^1^	1.82 × 10^2^
	Mean	1.07	3.22 × 10^3^	4.38	2.70 × 10⁻^1^	−4.61 × 10^4^	5.91 × 10^3^	1.45 × 10^1^	2.40 × 10^2^
	Std	2.67	1.81 × 10^3^	6.68	6.99 × 10⁻^1^	3.61 × 10^3^	4.63 × 10^3^	5.90 × 10⁻^1^	3.08 × 10^1^
	Speed	1.57 × 10^1^	1.65 × 10^1^	1.63 × 10^1^	1.61 × 10^1^	1.87 × 10^1^	1.75 × 10^1^	1.91 × 10^1^	1.78 × 10^1^
WOA	Best	0	6.03 × 10^2^	0	0	−8.38 × 10^4^	1.97 × 10^2^	0	0
	Mean	0	2.95 × 10^3^	0	0	−8.38 × 10^4^	1.97 × 10^2^	2.55 × 10⁻^15^	0
	Std	0	6.71 × 10^2^	0	0	8.94 × 10⁻^2^	1.27 × 10⁻^1^	2.41 × 10⁻^15^	0
	Speed	1.57 × 10^1^	1.55 × 10^1^	1.63 × 10^1^	1.49 × 10^1^	1.70 × 10^1^	1.63 × 10^1^	1.54 × 10^1^	1.53 × 10^1^

**Table 9 biomimetics-10-00222-t009:** Top algorithms evaluation for f9 to f15 and D = 200 (25 replications and 5000D FEs).

Alg.		f9	f10	f11	f12	f13	f14	f15
	Opt.	0	0	−1.40 × 10^2^	3.90 × 10^2^	−3.30 × 10^2^	−1.80 × 10^2^	0
ETOSO	Best	0	0	−1.40 × 10^2^	5.87 × 10^2^	−3.30 × 10^2^	−1.80 × 10^2^	0
	Mean	0	0	−1.40 × 10^2^	5.87 × 10^2^	−3.30 × 10^2^	−1.80 × 10^2^	0
	Std	0	0	2.32 × 10⁻^4^	1.46 × 10⁻^2^	8.72 × 10⁻^5^	2.12 × 10⁻^2^	0
	Speed	3.16	2.37	4.09	2.87	3.92	4.72	1.11 × 10^1^
DE	Best	3.55 × 10⁻^5^	4.81 × 10^1^	−1.36 × 10^2^	5.87 × 10^2^	6.47 × 10^1^	−1.80 × 10^2^	2.05 × 10⁻^1^
	Mean	4.32 × 10⁻^1^	4.76 × 10^6^	−1.33 × 10^2^	5.87 × 10^2^	2.24 × 10^2^	−1.80 × 10^2^	5.00 × 10⁻^1^
	Std	1.16	2.37 × 10^7^	2.39	6.46 × 10⁻^1^	6.30 × 10^1^	2.99 × 10⁻^1^	2.64 × 10⁻^1^
	Speed	1.65 × 10^1^	1.43 × 10^1^	1.91 × 10^1^	1.87 × 10^1^	1.87 × 10^1^	2.16 × 10^1^	5.75 × 10^1^
FPA	Best	2.90 × 10^3^	3.21 × 10^11^	−1.20 × 10^2^	3.22 × 10^7^	1.81 × 10^3^	2.57 × 10^3^	9.10 × 10^3^
	Mean	3.23 × 10^3^	3.60 × 10^11^	−1.19 × 10^2^	4.14 × 10^7^	2.36 × 10^3^	3.26 × 10^3^	1.06 × 10^4^
	Std	1.64 × 10^2^	1.79 × 10^10^	4.22 × 10⁻^1^	5.18 × 10^6^	1.78 × 10^2^	2.91 × 10^2^	7.08 × 10^2^
	Speed	3.63 × 10^1^	3.16 × 10^1^	3.80 × 10^1^	3.61 × 10^1^	3.76 × 10^1^	3.97 × 10^1^	1.21 × 10^2^
GWO	Best	0	0	−1.21 × 10^2^	1.49 × 10^7^	1.64 × 10^3^	1.35 × 10^3^	0
	Mean	0	0	−1.21 × 10^2^	1.76 × 10^7^	1.86 × 10^3^	1.72 × 10^3^	0
	Std	0	0	2.19 × 10⁻^1^	1.45 × 10^6^	1.13 × 10^2^	1.68 × 10^2^	0
	Speed	2.20 × 10^1^	2.10 × 10^1^	2.54 × 10^1^	2.09 × 10^1^	2.49 × 10^1^	3.85 × 10^1^	4.88 × 10^1^
HHO	Best	1.16 × 10⁻^5^	1.20 × 10^1^	−1.21 × 10^2^	6.11 × 10^2^	1.30 × 10^3^	−1.80 × 10^2^	6.22 × 10⁻^5^
	Mean	6.87 × 10⁻^3^	2.73 × 10^2^	−1.21 × 10^2^	9.60 × 10^3^	1.53 × 10^3^	−1.80 × 10^2^	1.19 × 10⁻^2^
	Std	2.61 × 10⁻^2^	2.96 × 10^2^	8.60 × 10⁻^2^	8.98 × 10^3^	1.08 × 10^2^	4.12 × 10⁻^2^	1.34 × 10⁻^2^
	Speed	5.43	1.78	3.95	2.44	3.56	7.22	3.54 × 10^1^
MFO	Best	1.80 × 10^2^	2.99 × 10^10^	−1.20 × 10^2^	2.07 × 10^7^	1.77 × 10^3^	2.10 × 10^3^	2.38 × 10^3^
	Mean	6.20 × 10^2^	5.56 × 10^10^	−1.20 × 10^2^	3.12 × 10^7^	2.32 × 10^3^	2.86 × 10^3^	4.28 × 10^3^
	Std	3.12 × 10^2^	2.02 × 10^10^	2.09 × 10⁻^1^	4.88 × 10^6^	2.56 × 10^2^	5.10 × 10^2^	9.37 × 10^2^
	Speed	2.50 × 10^1^	2.46 × 10^1^	2.91 × 10^1^	2.89 × 10^1^	2.88 × 10^1^	3.84 × 10^1^	3.73 × 10^1^
PDO	Best	0	0	−1.19 × 10^2^	5.01 × 10^7^	3.28 × 10^3^	5.41 × 10^3^	0
	Mean	0	0	−1.19 × 10^2^	5.02 × 10^7^	3.38 × 10^3^	5.42 × 10^3^	0
	Std	0	0	3.53 × 10⁻^2^	2.50 × 10^4^	6.16 × 10^1^	2.48	0
	Speed	1.79 × 10^1^	1.64 × 10^1^	2.21 × 10^1^	2.10 × 10^1^	2.14 × 10^1^	2.96 × 10^1^	2.71 × 10^1^
ROA	Best	0	0	−1.19 × 10^2^	4.42 × 10^7^	2.74 × 10^3^	4.67 × 10^3^	0
	Mean	0	0	−1.19 × 10^2^	4.64 × 10^7^	2.86 × 10^3^	5.01 × 10^3^	0
	Std	0	0	4.61 × 10⁻^2^	1.47 × 10^6^	7.66 × 10^1^	1.70 × 10^2^	0
	Speed	1.58 × 10^1^	1.62 × 10^1^	2.17 × 10^1^	2.02 × 10^1^	2.09 × 10^1^	2.90 × 10^1^	2.70 × 10^1^
SCA	Best	0	0	−1.20 × 10^2^	2.34 × 10^7^	2.29 × 10^3^	2.70 × 10^3^	0
	Mean	0	0	−1.19 × 10^2^	3.01 × 10^7^	2.53 × 10^3^	3.22 × 10^3^	0
	Std	0	0	6.54 × 10⁻^2^	2.40 × 10^6^	1.01 × 10^2^	3.37 × 10^2^	0
	Speed	1.65 × 10^1^	1.60 × 10^1^	2.09 × 10^1^	1.98 × 10^1^	2.03 × 10^1^	3.15 × 10^1^	2.76 × 10^1^
TLBO	Best	1.20 × 10⁻^4^	2.83 × 10^3^	−1.21 × 10^2^	6.32 × 10^2^	1.21 × 10^3^	−1.76 × 10^2^	7.72 × 10⁻^1^
	Mean	3.23 × 10⁻^2^	9.31 × 10^5^	−1.21 × 10^2^	7.98 × 10^4^	1.36 × 10^3^	−1.50 × 10^2^	1.03
	Std	4.27 × 10⁻^2^	4.01 × 10^6^	1.66 × 10⁻^1^	8.21 × 10^4^	9.44 × 10^1^	1.77 × 10^1^	1.59 × 10⁻^1^
	Speed	1.74 × 10^1^	1.70 × 10^1^	2.31 × 10^1^	2.11 × 10^1^	2.17 × 10^1^	3.08 × 10^1^	2.92 × 10^1^
WOA	Best	0	0	−1.19 × 10^2^	4.27 × 10^7^	2.76 × 10^3^	4.39 × 10^3^	0
	Mean	0	0	−1.19 × 10^2^	4.58 × 10^7^	2.99 × 10^3^	4.77 × 10^3^	8.88 × 10⁻^18^
	Std	0	0	3.52 × 10⁻^2^	1.27 × 10^6^	7.11 × 10^1^	1.74 × 10^2^	3.07 × 10⁻^17^
	Speed	1.56 × 10^1^	1.70 × 10^1^	2.19 × 10^1^	2.13 × 10^1^	2.07 × 10^1^	3.05 × 10^1^	2.55 × 10^1^

**Table 10 biomimetics-10-00222-t010:** Ranking of top algorithms in terms of performance (P), speed (S), and consistency (C).

Alg.	D = 30			D = 50			D = 100			D = 200		
	P	S	C	P	S	C	P	S	C	P	S	C
ETOSO	1	2	1	1	2	1	1	2	1	1	2	1
DE	2	4	6	4	4	5	4	4	6	6	3	8
FPA	10	11	10	10	11	10	11	11	10	11	11	10
GWO	3	3	4	2	3	6	2	3	7	2	9	5
HHO	8	1	8	6	1	8	6	1	8	3	1	7
MFO	11	10	11	11	10	11	10	10	11	10	10	11
PDO	9	9	2	9	9	3	9	9	2	9	7	3
ROA	4	6	3	3	6	2	3	6	3	4	5	2
SCA	5	5	5	4	5	4	4	5	4	4	4	6
TLBO	7	8	8	7	8	9	8	8	9	8	8	9
WOA	6	7	7	8	7	7	7	7	5	6	6	4

**Table 11 biomimetics-10-00222-t011:** *p*-values from Wilcoxon signed-rank test with D = 50 and ps = 10 (using 5000D functions evaluations and 25 replications).

Algorithm	ETOSO	DE	FPA	GWO	HHO	MFO	PDO	ROA	SCA	TLBO	WOA
f1	1	9.73 × 10⁻^11^	9.73 × 10⁻^11^	1	9.73 × 10⁻^11^	9.73 × 10⁻^11^	1	1	1	9.73 × 10⁻^11^	1
f2	1	9.73 × 10⁻^11^	9.73 × 10⁻^11^	0	9.73 × 10⁻^11^	9.73 × 10⁻^11^	1	1	9.73 × 10⁻^11^	9.73 × 10⁻^11^	9.73 × 10⁻^11^
f3	1	9.73 × 10⁻^11^	9.73 × 10⁻^11^	1	9.73 × 10⁻^11^	9.73 × 10⁻^11^	1	1	1	9.73 × 10⁻^11^	1
f4	1	9.73 × 10⁻^11^	9.73 × 10⁻^11^	1	9.73 × 10⁻^11^	9.73 × 10⁻^11^	1	1	1	9.73 × 10⁻^11^	1
f5	1	1.42 × 10⁻^9^	1.42 × 10⁻^9^	1.42 × 10⁻^9^	1.42 × 10⁻^9^	1.42 × 10⁻^9^	1.42 × 10⁻^9^	4.25 × 10⁻^6^	2.57 × 10⁻^8^	1.42 × 10⁻^9^	3.67 × 10⁻^9^
f6	1	2.57 × 10⁻^8^	1.42 × 10⁻^9^	2.44 × 10⁻^2^	1.42 × 10⁻^9^	4.26 × 10⁻^6^	1.42 × 10⁻^9^	1.42 × 10⁻^9^	1.42 × 10⁻^9^	1.42 × 10⁻^9^	1.42 × 10⁻^9^
f7	1	9.65 × 10⁻^11^	9.73 × 10⁻^11^	9.61 × 10⁻^11^	9.73 × 10⁻^11^	9.73 × 10⁻^11^	1	1	1	2.33 × 10⁻^11^	2.08 × 10⁻^2^
f8	1	9.73 × 10⁻^11^	9.73 × 10⁻^11^	1	9.73 × 10⁻^11^	9.73 × 10⁻^11^	1	1	1	9.73 × 10⁻^11^	1
f9	1	9.73 × 10⁻^11^	9.73 × 10⁻^11^	1	9.73 × 10⁻^11^	9.73 × 10⁻^11^	1	1	1	9.73 × 10⁻^11^	3.37 × 10⁻^1^
f10	1	9.73 × 10⁻^11^	9.73 × 10⁻^11^	1	9.73 × 10⁻^11^	9.73 × 10⁻^11^	1	1	1	9.73 × 10⁻^11^	0.00
f11	1	1.42 × 10⁻^9^	1.42 × 10⁻^9^	1.42 × 10⁻^9^	1.42 × 10⁻^9^	1.42 × 10⁻^9^	1.42 × 10⁻^9^	1.42 × 10⁻^9^	1.42 × 10⁻^9^	1.42 × 10⁻^9^	1.42 × 10⁻^9^
f12	1	1.41 × 10⁻^9^	3.88 × 10⁻^5^	1.41 × 10⁻^9^	2.56 × 10⁻^8^	1.41 × 10⁻^9^	1.41 × 10⁻^9^	1.41 × 10⁻^9^	1.41 × 10⁻^9^	1.41 × 10⁻^9^	1.41 × 10⁻^9^
f13	1	1.42 × 10⁻^9^	1.42 × 10⁻^9^	1.42 × 10⁻^9^	1.42 × 10⁻^9^	1.42 × 10⁻^9^	1.42 × 10⁻^9^	1.42 × 10⁻^9^	1.42 × 10⁻^9^	1.42 × 10⁻^9^	1.42 × 10⁻^9^
f14	1	9.18 × 10⁻^10^	1.31 × 10⁻^6^	9.18 × 10⁻^10^	9.18 × 10⁻^10^	9.18 × 10⁻^10^	9.18 × 10⁻^10^	9.18 × 10⁻^10^	9.18 × 10⁻^10^	9.18 × 10⁻^10^	9.18 × 10⁻^10^
f15	1	9.73 × 10⁻^11^	9.73 × 10⁻^11^	1	9.73 × 10⁻^11^	9.73 × 10⁻^11^	1	1	1	1.62 × 10⁻^1^	9.73 × 10⁻^11^

**Table 12 biomimetics-10-00222-t012:** Wilcoxon significance table with D = 50 and ps = 10 (using 5000D functions evaluations and 25 replications).

Algorithm	ETOSO	DE	FPA	GWO	HHO	MFO	PDO	ROA	SCA	TLBO	WOA
f1	0	1	1	0	1	1	0	0	0	1	0
f2	0	1	1	1	1	1	0	0	1	1	1
f3	0	1	1	0	1	1	0	0	0	1	0
f4	0	1	1	0	1	1	0	0	0	1	0
f5	0	1	1	1	1	1	1	1	1	1	1
f6	0	1	1	1	1	1	1	1	1	1	1
f7	0	1	1	1	1	1	0	0	0	1	1
f8	0	1	1	0	1	1	0	0	0	1	0
f9	0	1	1	0	1	1	0	0	0	1	0
f10	0	1	1	0	1	1	0	0	0	1	1
f11	0	1	1	1	1	1	1	1	1	1	1
f12	0	1	1	1	1	1	1	1	1	1	1
f13	0	1	1	1	1	1	1	1	1	1	1
f14	0	1	1	1	1	1	1	1	1	1	1
f15	0	1	1	0	1	1	0	0	0	1	0

**Table 13 biomimetics-10-00222-t013:** Cliff’s Delta table with D = 50 and ps = 10 (using 5000D functions evaluations and 25 replications).

Algorithm	ETOSO	DE	FPA	GWO	HHO	MFO	PDO	ROA	SCA	TLBO	WOA
f1	0	−1	−1	0	−1	−1	0	0	0	−1	0
f2	0	−1	−1	−1	−1	−1	0	0	−1	−1	−1
f3	0	−1	−1	0	−1	−1	0	0	0	−1	0
f4	0	−1	−1	0	−1	−1	0	0	0	−1	0
f5	0	−1	−1	−1	−1	−1	−1	−0.76	−0.92	−1	−0.9744
f6	0	−0.92	−1	−0.3728	−1	−0.76	−1	1	−1	−1	−1
f7	0	−1	−1	−1	−1	−1	0	0	0	−1	−0.2
f8	0	−1	−1	0	−1	−1	0	0	0	−1	0
f9	0	−1	−1	0	−1	−1	0	0	0	−1	−0.04
f10	0	−1	−1	0	−1	−1	0	0	0	−1	0
f11	0	−1	−1	−1	−1	−1	−1	−1	−1	−1	−1
f12	0	−1	−0.68	−1	−0.92	−1	−1	−1	−1	−1	−1
f13	0	−1	−1	−1	−1	−1	−1	−1	−1	−1	−1
f14	0	−1	−0.7872	−1	−1	−1	−1	−1	−1	−1	−1
f15	0	−1	−1	0	−1	−1	0	0	−0.08	−1	0

**Table 14 biomimetics-10-00222-t014:** Performance metrics of optimization algorithms with D = 50 and ps = 10 (using 5000D functions evaluations and 25 replications).

Algorithm	Perfect Hits	Closest to Min	Std Dev ≤ Threshold	NAE	TNAE	Avg Speed (Sec)
ETOSO	9	9	14	1.87 × 10^3^	4.03	8.52 × 10⁻^1^
DE	0	1	1	3.82 × 10^6^	3.25 × 10^3^	4.20
FPA	0	1	1	1.35 × 10^21^	3.25 × 10^9^	5.47
GWO	7	7	8	1.96 × 10^3^	5.31	1.97
HHO	0	1	1	5.74 × 10^3^	2.08 × 10^3^	7.74 × 10⁻^1^
MFO	0	1	0	1.04 × 10^9^	2.56 × 10^8^	4.71
PDO	9	9	9	2.18 × 10^3^	4.37	4.45
ROA	9	9	10^1^	2.10 × 10^3^	8.53 × 10⁻^1^	8.36
SCA	7	7	8	2.01 × 10^8^	9.58 × 10^3^	4.10
TLBO	0	1	1	7.99 × 10^3^	5.52 × 10^3^	4.30
WOA	6	6	8	1.08 × 10^4^	4.83	4.15

**Table 15 biomimetics-10-00222-t015:** Comparative analysis of algorithm computational complexity and overhead.

Algorithm	Computational Complexity	Overhead Category	Key Operations Contributing to Overhead
ETOSO	O[FE⋅(D + log (ps))]	Moderate	Moderate overhead due to efficient exploitation mechanisms, including sorting.
HHO	O(FE⋅D)	Low	Avoids expensive mathematical operations; relies on simple arithmetic and conditional logic.
DE	O(FE⋅D)	Moderate	Involves mutation and crossover operations.
TLBO	O(FE⋅D)	Moderate	Requires mean calculations and duplicated loops.
GWO	O(FE⋅D)	Moderate	Involves sorting and best candidate calculations.
WOA	O(FE⋅D)	High	Uses trigonometric and exponential functions in spiral updates.
MFO	O(FE⋅D)	High	Relies on logarithmic spiral calculations.
SCA	O(FE⋅D)	High	Uses trigonometric functions for position updates.
FPA	O(FE⋅D)	Very High	Involves Levy flight calculations.
ROA	O(FE⋅D)	Very High	Combines exponential and trigonometric functions with host-switching logic.
PDO	O(FE⋅D)	Moderate	Chaotic tent initialization and loop-based positional updates.

## Data Availability

Data is contained within the article.

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
