# Peer review of "An Enhanced Team-Oriented Swarm Optimization Algorithm (ETOSO) for Robust and Efficient High-Dimensional Search"

_biomimetics, 2025, doi:10.3390/biomimetics10040222_

Round 1
Reviewer 1 Report
Comments and Suggestions for Authors
In this work, authors present an enhancement of the Team-Oriented Swarm Optimization (TOSO) algorithm, named Enhanced Team-Oriented Swarm Optimization (ETOSO). The improvement of the algorithm was structured by improving exploration and exploitation mechanisms. The superiority of the new ETOSO has been proved through a comprehensive comparative evaluation against 26 established nature-inspired optimization algorithms across 15 benchmark functions using several dimensions. Statistical evaluations, including mean performance and standard deviation, are well-documented. The paper provides a clear and structured explanation of the new ETOSO algorithm, with mathematical formulations and pseudocode.
From my point of view, I recommend the acceptance of publishing this work in "Biomimetics" after taking the following comments into consideration:
1. Although the work mostly concentrates on benchmark functions, it does not apply to real optimization cases. The practical relevance would be increased by adding a real-world case study (such as engineering design or machine learning parameter tuning).
2. The idea that ETOSO is parameter-free would be strengthened by a thorough examination of its performance in various scenarios, such as those with differing swarm sizes and problem complexity.
3. A discussion on the computational complexity of ETOSO compared to other algorithms is missing. This would help readers assess its scalability for large-scale problems.
Reviewer 2 Report
Comments and Suggestions for Authors
Authors in the paper proposed An Enhanced Team-Oriented Swarm Optimization Algorithm (ETOSO) for Robust and Efficient High-Dimensional Search. The presentation of the paper, as well as the organization of the results, is highly disorganized. Furthermore, the most critical issue is that the author attempts to enhance an optimization algorithm that has been rarely used, with only seven citations since 2013. This particular algorithm has not gained widespread acceptance within the scientific community.
In order to give readers a better understanding of the current research status of the issue, which this paper addresses, I suggest expanding this section with related works.
The literature review section should focus on reviewing hybrid or improved metaheuristic algorithms rather than the original version of the algorithm. Therefore, Section 2 needs to be rewritten accordingly.
There was no declaration of abbreviations used by the authors.
Provide a brief overview of the method's limitations and practical benefits.
I suggest that the proposed model be compared with more Benchmark Functions that the author mention just 15 of 23 well-known function from CEC2014-2015. I strongly suggest to add CEC2017 and CEC2019 with convergence curve and boxplot analysis.
Also in statistical analysis also add Wilcoxon test analyze.
A number of minor issues have been identified. It is necessary for the authors to review the entire paper in order to correct these errors.
The proposed algorithm should be discussed in light of future work and limitations.
Reviewer 3 Report
Comments and Suggestions for Authors
- Page 3 - Line 95: The Bat Algorithm and the Bees Algorithm share the same abbreviation (BA). Please make the necessary adjustments.
- General Comment - Introduction: Since this paper is not a review paper, the extensive description of metaheuristic algorithms should be drastically shorten.
Round 2
Reviewer 2 Report
Comments and Suggestions for Authors
It can be accepted in this form.